# AIR: Rethinking Image-Text Offset Alignment for Zero-Shot Generative Model Adaptation

## Abstract

Zero-shot generative model adaptation (ZSGM) aims to adapt pre-trained generative models using only textual descriptions. ZSGM is particularly valuable for data-scarce target domains, such as rare concepts or artistic styles, where obtaining training samples is challenging. Central to all existing ZSGM methods is the foundational assumption that image-text offsets in CLIP's multimodal representation space are well aligned to guide adaptation. **In this work**, we present two main contributions. First, we question this foundational assumption by conducting the first comprehensive empirical analysis of image-text offset alignment in CLIP space within the ZSGM context. Our findings reveal not only noticeable misalignment but also a meaningful positive correlation between image-text offset misalignment and concept distance across six large datasets and four multimodal spaces. Second, leveraging this discovery, we propose Adaptation with Iterative Refinement (AIR), the first method focused on improving sample quality for ZSGM. Our method iteratively refines text offsets and reduces image-text offset misalignment, using anchor sampling and a novel prompt learning approach. Comprehensive experiments accross **32** experiment setups, including qualitative, quantitative, and user studies, consistently show that AIR achieves state-of-the-art performance. **Code and additional experiments are available in the supplementary material.**

## 1 Introduction

Generative models, including Generative Adversarial Networks (GANs) (Goodfellow et al., 2014; Karras et al., 2020b; Kang et al., 2023; Huang et al., 2024) and Diffusion Models (Rombach et al., 2022; Peebles & Xie, 2023; Esser et al., 2024), have made significant strides in producing high-fidelity and diverse images. However, their training requires extensive datasets, such as 70K images for StyleGAN2 (Karras et al., 2020c) or 400M for Stable Diffusion (Rombach et al., 2022), which are often unavailable in data-scarce domains like rare species, rare concepts, or artistic styles. Training with limited data frequently results in mode collapse (Abdollahzadeh et al., 2023), emphasizing the need to address these challenges.

**Generative model adaptation** has become a key research area, leveraging pre-trained generators from rich source domains to adapt to data-scarce target domains(Li et al., 2020; Ojha et al., 2021; Zhao et al., 2022b; Zhou et al., 2024; Anees et al., 2024; Zhu et al., 2025; Guo et al., 2025; Cai et al., 2025). This approach exploits the diversity of source models to generate robust, varied samples for rare or limited-data domains, improving both diversity and quality.

**Zero-shot generative model adaptation (ZSGM)** represents a significant advancement in this field, relying exclusively on textual descriptions without target images. NADA (Gal et al., 2022), a pioneering effort, utilizes text offsets within CLIP multimodal embedding space to guide adaptation by aligning image offsets (from source to adapted generator) with text offsets (from source to target). This method is grounded in the core assumption that the offset between image embeddings and the offset between their corresponding text embeddings in CLIP space are well aligned. This image-text offset alignment has been the foundational assumption of all subsequent ZSGM approaches (Guo et al., 2023; Jeon et al., 2023). See Sec. M for detailed discussion on related work.

**In this work**, we challenge the foundational assumption underlying ZSGM. For the first time in literature, we conduct a comprehensive empirical analysis of offset alignment within CLIP embedding

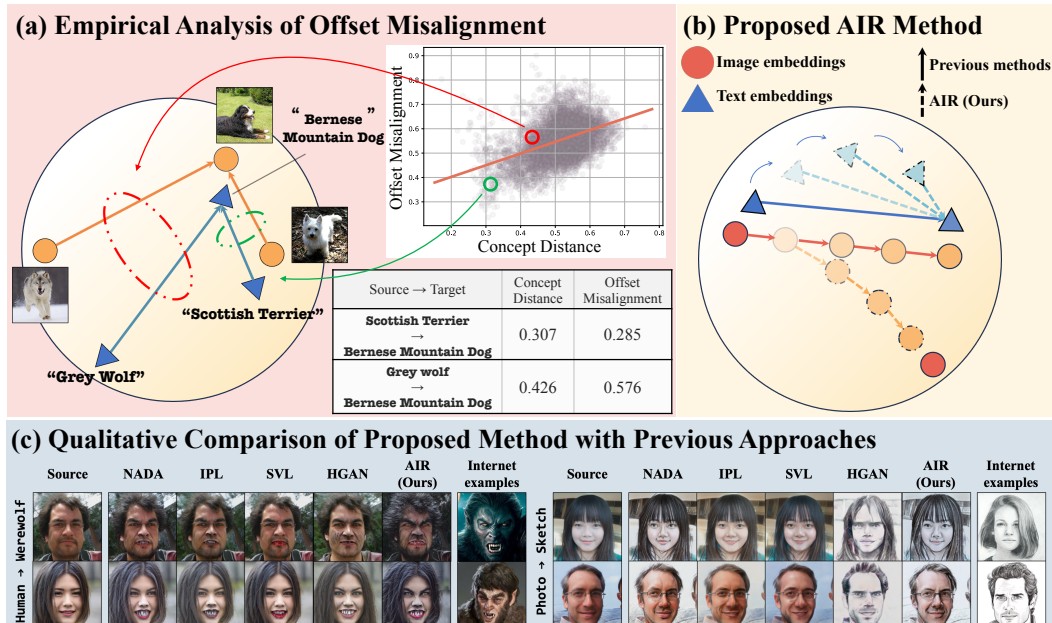

Figure 1: **Our contributions: (a)** We question the core assumption of all existing ZSGM methods: image-text offset alignment. We perform a comprehensive analysis of offset alignment in CLIP embedding space. **Our analysis reveals that not only there is noticeable misalignment between image offset (orange arrow) and text offset (blue arrow) but also a meaningful positive correlation between offset misalignment and concept distance**. For example, in the ImageNet-1K dataset, the "Grey Wolf" is a more distant concept to the "Bernese Mountain Dog" (concept distance=0.426) than the "Scottish Terrier" (concept distance=0.307). Accordingly, "Grey Wolf" → "Bernese Mountain Dog" has higher offset misalignment than "Scottish Terrier" → "Bernese Mountain Dog" (0.576 vs 0.285). This misalignment is overlooked in existing approaches, resulting in degradation in target domain image quality (Sec. 3). **(b)** Leveraging our discovery, we propose Adaptation with Iterative Refinement (AIR) to iteratively refine text offsets and reduce offset misalignment for ZSGM (Sec. 4). **(c)** Our proposed AIR consistently achieves improved quality across diverse setups by better capturing rich target domain's style and details. (see Sec. 5 and Sec. A for detailed results).

space in the context of ZSGM. **Our primary findings** reveal not only the presence of noticeable misalignment in some instances but also a meaningful positive correlation between image-text offset misalignment and concept distance: for closely related concept pairs, misalignment tends to be smaller, whereas it increases as concepts become more distant (Fig. 1). This correlation is consistently observed in six large public datasets and four contrastive learning-based multimodal spaces.

Leveraging our discovery of the positive correlation between misalignment and concept distance, we propose Adaptation with Iterative Refinement (AIR), a novel framework designed to enhance the quality of generated images in ZSGM. After limited iterations of initial adaptation, the adapted generator encodes a concept closer to the target than the source (Sec. H), potentially reducing image-text offset misalignment. Building on this insight, AIR iteratively samples intermediate adapted generators as anchors during the adaptation, refining offsets with these anchors to enhance guidance accuracy. As the textual descriptions of these anchor are unknown, we introduce a novel prompt learning strategy to infer them dynamically. Our extensive results consistently demonstrate improved adaptation quality across diverse setups. Our main contributions are summarized as follows:

- We challenge the foundational assumption of ZSGM by conducting the first comprehensive empirical analysis of offset alignment within CLIP embedding space, revealing noticeable misalignment and a meaningful positive correlation with concept distance (Sec. 3).

- We introduce Adaptation with Iterative Refinement (AIR), a novel framework that leverages the discovered correlation to improve generated image quality in zero-shot adaptation, utilizing iterative anchor sampling and a new prompt learning strategy to dynamically infer unknown textual descriptions (Sec. 4).

- Our extensive experiments across 32 diverse setups, including the first application to diffusion models, demonstrate consistent improvements in adaptation quality, validated by qualitative, quantitative, and user study results, achieving state-of-the-art performance (Sec. 5 and Supp.).

**Remark:** Our discovery of *image-text offset misalignment* in CLIP multimodal space can be viewed as an analogy to *text offset misalignment* studies in unimodal text embedding space for natural language processing (NLP). In NLP, *analogical reasoning* (Mikolov et al., 2013c;a;b; Levy & Goldberg, 2014) relies on aligning offsets between word vectors, such as alignment between $E_v(\text{"Man"})$ - $E_v(\text{"Woman"})$, and $E_v(\text{"King"})$ - $E_v(\text{"Queen"})$, where $E_v$ denotes a text vector. Research indicates that the accuracy of analogical reasoning improves with similar, nearby concepts but decreases with growing distance (Levy et al., 2015; Köper et al., 2015; Rogers et al., 2017; Fournier et al., 2020). Similarly, our finding of a positive correlation between image-text offset misalignment and concept distance in CLIP reveals a similar distance-dependent relationship in CLIP space.

## 2 PRELIMINARIES: DIRECTIONAL CLIP LOSS

In zero-shot generative model adaptation setup (Gal et al., 2022), given a pre-trained generator $G_{\mathcal{S}}$ on the source domain $\mathcal{S}$, and textual descriptions of source and target domains, denoted by $T_{\mathcal{S}}$ and $T_{\mathcal{T}}$ respectively, the goal is to shift $G_{\mathcal{S}}$ to target domain $\mathcal{T}$ to generate diverse and high-quality images from this domain (Abdollahzadeh et al., 2023). For this adaptation, current approaches (Gal et al., 2022; Guo et al., 2023; Jeon et al., 2023) use the CLIP model (Radford et al., 2021) as the source of supervision, and assume that text and image offsets (between $\mathcal{S}$ and $\mathcal{T}$) are well-aligned in CLIP representation space. Therefore, the text offset is computed based on the provided textual descriptions of the source and target. Then, the trainable generator is initialized with the parameters of the $G_{\mathcal{S}}$, and optimized in a way to align image offset with text offset, leading to the directional CLIP loss:

$$\mathcal{L}_{direction} = 1 - \cos(\Delta I_{\mathcal{S} \to t}, \Delta T_{\mathcal{S} \to \mathcal{T}}),$$
$$\text{where } \Delta I_{\mathcal{S} \to t} = E_I(G_t(w)) - E_I(G_{\mathcal{S}}(w)), \tag{1}$$
$$\text{and } \Delta T_{\mathcal{S} \to \mathcal{T}} = E_T(T_{\mathcal{T}}) - E_T(T_{\mathcal{S}})$$

where $\cos(x, y) = x \cdot y / |x||y|$ represents the cosine similarity. $E_T$ and $E_I$ denote the CLIP text and image encoders, respectively. $G_t$ denotes the trainable generator in iteration $t$ of adaptation. $\Delta I_{\mathcal{S} \to t}$ denotes the image offset computed from the source generator to the trainable generator, and $\Delta T_{\mathcal{S} \to \mathcal{T}}$ denotes the text offset from source to target.

## 3 A CLOSER LOOK AT OFFSET MISALIGNMENT IN CLIP SPACE

Previous works assume that for two different concepts $\alpha$ and $\beta$, the image offset $\Delta I_{\alpha \to \beta}$ and text offset $\Delta T_{\alpha \to \beta}$ are well aligned in the multimodal CLIP embedding space. This alignment assumption underlies the directional loss (Eq. 1). We postulate that this assumption has two major limitations:

- CLIP (Radford et al., 2021) is trained with contrastive loss to maximize cosine similarity between corresponding image-text pairs, *i.e.,* maximize $\cos(E_I(I_\alpha), E_T(T_\alpha))$ for concept $\alpha$ (*e.g.*, cat), or maximize $\cos(E_I(I_\beta), E_T(T_\beta))$ for concept $\beta$ (*e.g.*, dog). Note that the degree of alignment of image offset $\Delta I_{\alpha \to \beta}$ and text offset $\Delta T_{\alpha \to \beta}$ in CLIP space is not studied in the literature.

- In addition, this degree of alignment between $\Delta I_{\alpha \to \beta}$ and $\Delta T_{\alpha \to \beta}$ may vary based on the distance between two concepts $\alpha$ and $\beta$.

In this section, we take a closer look at this degree of offset alignment between two different modalities in CLIP space. First, inspired by offset misalignment in NLP, we conduct an empirical study on large public datasets to analyze the offset misalignment between image and text modalities in CLIP embedding space. Our analysis suggests that **there is a misalignment between $\Delta I_{\alpha \to \beta}$ and $\Delta T_{\alpha \to \beta}$ in CLIP embedding space, and this misalignment increases as concepts $\alpha$ and $\beta$ become more distant**. Second, we take a further step and design an experiment to evaluate the effect of this offset misalignment in generative model adaptation using directional loss (Eq. 1). Our experimental results suggest that **less offset misalignment in CLIP embedding space leads to a better generative model adaptation with directional loss**.

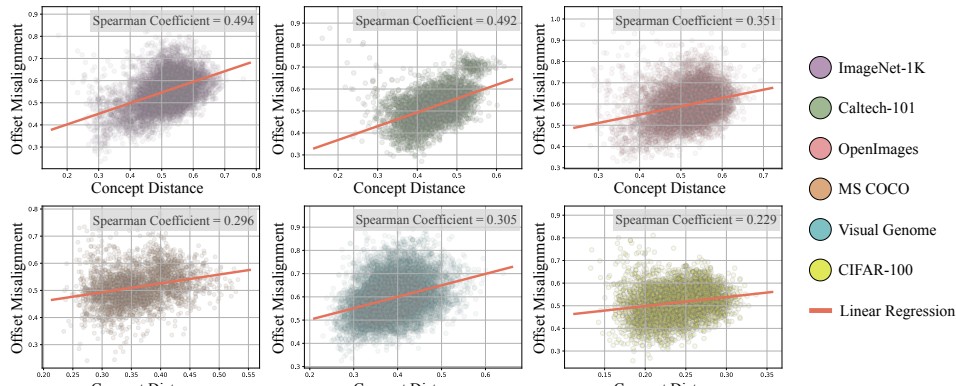

Figure 2: **Empirical analysis of offset misalignment in ViT-based CLIP space:** We plot the offset misalignment (Eq. 2) vs concept distance for $N = 5000$ of text-image pairs in CLIP space which are sampled from 6 large publicly available datasets (details in Sec. C.1; total $30,000$ text-image pairs). Our results show that there is a meaningful correlation (measured by Spearman's coefficient (Zar, 2005)) between offset misalignment and concept distance for datasets with different distributions, i.e., close concepts has less offset misalignment. **Furthermore, we have consistent observations for three additional CLIP-like representation spaces, see Sec. G.**

### 3.1 EMPIRICAL ANALYSIS OF OFFSET MISALIGNMENT

In this section, we conduct an empirical experiment on public datasets to evaluate the degree of alignment between image and text offsets. We randomly sample two classes for each dataset as a pair of concept $(\alpha, \beta)$. Then, the images within each class are used alongside the related textual description (*e.g.,* label) of each class to measure offset misalignment $\mathcal{M}(\alpha, \beta)$ in a similar way to directional loss:

$$\mathcal{M}(\alpha, \beta) = 1 - \cos(\Delta I_{\alpha \to \beta}, \Delta T_{\alpha \to \beta}),$$
$$\text{where } \Delta I_{\alpha \to \beta} = E_I \overline{(I_\beta)} - E_I \overline{(I_\alpha)}, \tag{2}$$
$$\text{and } \Delta T_{\alpha \to \beta} = E_T(T_\beta) - E_T(T_\alpha)$$

where $E_I \overline{(I_\alpha)}$ is the average embedding of all images of the class (concept) $\alpha$ in CLIP space. To measure the distance between two concepts denoted by $\mathcal{D}(\alpha, \beta)$, we use cosine similarity between images of two classes, *i.e.,* $\mathcal{D}(\alpha, \beta) = 1 - \cos(E_I \overline{(I_\beta)}, E_I \overline{(I_\alpha)})$. We repeat this for $N = 5000$ pairs of concepts for each dataset. Then, we plot $\mathcal{M}(\alpha, \beta)$ against $\mathcal{D}(\alpha, \beta)$ for each pair of concepts.

**Experimental Setup.** In this experiment, we use CLIP ViT-B/32 as vision encoder. We use 6 large and multi-class datasets that are publicly available, including ImageNet (Deng et al., 2009), Caltech-101 (Fei-Fei et al., 2007), OpenImages (Kuznetsova et al., 2020), COCO (Lin et al., 2014), Visual Genome (Krishna et al., 2017), and CIFAR-100 (Krizhevsky et al., 2009) (details in Sec. C.1).

**Results.** Fig. 2 shows the offset misalignment against the concept distance for $N = 5000$ pairs of concepts for 6 public datasets. As shown in the plots, for all datasets, apart from their different distributions and characteristics, there is a positive correlation between offset misalignment and concept distance. Particularly, if two concepts $\alpha$ and $\beta$ are distant, there is a higher misalignment between image offset $\Delta I_{\alpha \to \beta}$ and corresponding text offset $\Delta T_{\alpha \to \beta}$. This means that given $I_\alpha, T_\alpha$ and $T_\beta$, it is sub-optimal to align $\Delta I_{\alpha \to \beta}$ and $\Delta T_{\alpha \to \beta}$ to find $I_\beta$. On the other hand, if two concepts $\alpha$ and $\beta$ are closer, potentially, it is more accurate to align $\Delta I_{\alpha \to \beta}$ and $\Delta T_{\alpha \to \beta}$ to find $I_\beta$.

**Remark:** Our work is the first to reveal that **offset misalignment between image and text modalities in CLIP correlates positively with concept distance**. In what follows, we design an experiment to show that less offset misalignment leads to a better generative adaptation with directional loss.

### 3.2 IMPACT OF OFFSET MISALIGNMENT ON GENERATIVE MODEL ADAPTATION

In the previous section, we performed an empirical study that revealed the offset misalignment. In this section, we take a step further and investigate the effect of this misalignment on the generative

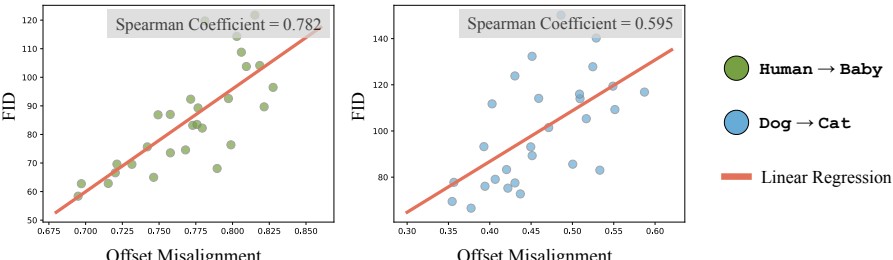

Figure 3: **Impact of offset misalignment on zero-shot generative model adaptation with directional loss:** For each of the two setups, we fix the source domain and augment the text description of the target domain to simulate various degrees of misalignment between image offset and text offset. Then, we perform the adaptation using directional loss in Eq. 1 for each setup. Results show that adaptation performance degrades by increasing the offset misalignment.

model adaptation from a source domain (concept) $\mathcal{S}$ to a target domain (concept) $\mathcal{T}$. Specifically, following ZSGM setup (Gal et al., 2022), for source domain $\mathcal{S}$, we assume a pre-trained generator $G_\mathcal{S}$ and a text description $T_\mathcal{S}$ is available. However, for the target domain, only text description $T_\mathcal{T}$ is available. To simulate different degrees of misalignment between source and target, we augment target text to get a set of text descriptions $\{T_\mathcal{T}^i\}$. Then, we perform zero-shot adaptation using the directional loss (Eq. 1) from the source domain $\mathcal{S}$ to each of these target text $T_\mathcal{T}^i$ and measure the generation performance of adapted generator.

**Experimental Setup.** For this experiment, we perform adaptation on Human → Baby and Dog → Cat. We use StyleGAN2-ADA (Karras et al., 2020a) pre-trained on FFHQ (Karras et al., 2019) and AFHQ-Dog (Choi et al., 2020) as the pre-trained model. We fix the source text $T_\mathcal{S}$ and augment the target text $T_\mathcal{T}$ by sampling handcrafted prompts from the CLIP ImageNet template (INt)[1] in order to simulate different degrees of misalignment (see Sec. C.2). Then, we follow exactly the same hyperparameters as NADA (see Sec. C.3) to adapt the source generator to different target text $T_\mathcal{T}^i$. We use FID to measure the performance of the adapted generator against offset misalignment.

**Our results** in Fig. 3 demonstrates that in general, **increasing the offset misalignment degrades the performance of the zero-shot generative adaption with directional loss**. Motivated by this finding, we propose an approach to iteratively refine the adaptation direction.

# 4 METHODOLOGY: ADAPTATION WITH ITERATIVE REFINEMENT

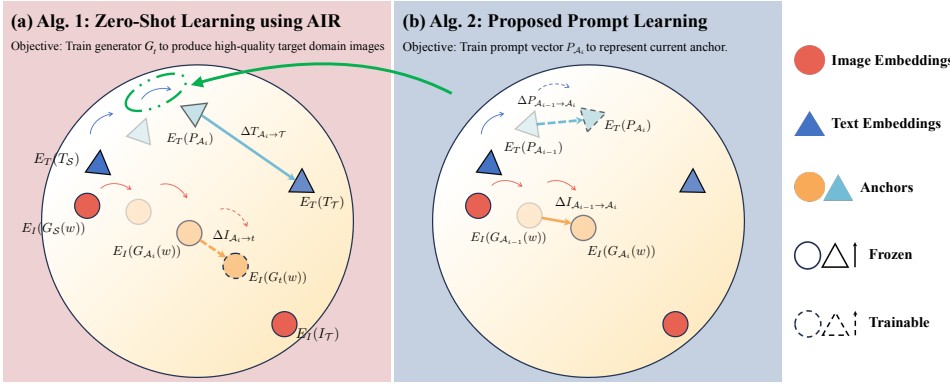

Figure 4: **Illustration of the proposed AIR method:** (a) Zero-shot learning scheme using Adaptation with Iterative Refinement (AIR) (Sec. 4.1). (b) The proposed prompt learning method to learn text embedding $P_{\mathcal{A}_i}$ for the anchor $\mathcal{A}_i$ (Sec. 4.2).

Our analysis in Sec. 3 suggests that closer concepts tend to have less offset misalignment in CLIP space, resulting in a more accurate directional loss (Eq. 1) for adaptation. Here, we leverage this property to enhance the zero-shot generative model adaptation with directional loss.

---

[1] https://github.com/openai/CLIP/blob/main/notebooks/Prompt_Engineering_for_ImageNet.ipynb

Specifically, even though the concept distance between source $\mathcal{S}$ and target $\mathcal{T}$ is fixed, after a limited number of initial adaptation iterations using directional loss, the encoded concept in the adapted generator is already closer to the target domain than the encoded concept in source generator (see Sec. H ). For example, when adapting a generator pre-trained on `Photo` to the target domain `Painting`, after limited adaptation iterations, the adapted generator already encodes more knowledge related to the `Painting` domain than the pre-trained generator.

Therefore, we use the adapted generator as the new anchor (denoted by $G_{\mathcal{A}}$), and compute the directional loss from this anchor point to the target. We update this anchor point iteratively during adaptation, as we move closer to the target domain. Because of the smaller concept distance, our previous analysis suggests that the directional loss computed based on $G_{\mathcal{A}}$ can provide better guidance, and this improves the adaptation direction solely computed based on $G_{\mathcal{S}}$. One challenge of using $G_{\mathcal{A}}$ for directional loss is that the corresponding text prompt $P_{\mathcal{A}}$ that describes this concept is unknown. In what follows, first, we discuss the details of the proposed *Adaptation with Iterative Refinement (AIR)* in Sec. 4.1. Then, to infer the unknown $P_{\mathcal{A}}$ within the directional loss of AIR, we introduce a prompt learning method in Sec. 4.2.

## 4.1 ADAPTATION WITH ITERATIVE REFINEMENT (AIR)

In our proposed approach, first, we adapt the generator to the target domain for $t_{thresh}$ iterations using directional loss in Eq. 1 to make sure the adapted generator has moved closer to the target domain. Then, in each $t_{int}$ interval of adaptation, we sample the adapted generator as the new anchor point. [2] We denote $i^{th}$ sampled anchor by $G_{\mathcal{A}_i}$. **To reduce offset misalignment and provide more accurate direction, we use the anchor point $\mathcal{A}_i$ instead of source point $\mathcal{S}$ for computing the directional loss.** The proposed AIR scheme is illustrated in Fig. 4 (a). The image offset with anchor point $\mathcal{A}_i$ is computed based on the sampled generator $G_{\mathcal{A}_i}$, and the trainable generator $G_t$: $\Delta I_{\mathcal{A}_i \to t} = E_I(G_t(w)) - E_I(G_{\mathcal{A}_i}(w))$. Assuming that the anchor point is described by the prompt $P_{\mathcal{A}_i}$ in the text domain (details of inferring $P_{\mathcal{A}_i}$ will be discussed in Sec. 4.2), the text offset with anchor point is calculated as follows: $\Delta T_{\mathcal{A}_i \to \mathcal{T}} = E_T(T_{\mathcal{T}}) - E_T(P_{\mathcal{A}_i})$. Finally, the adaptive loss $\mathcal{L}_{adaptive}$ is computed by aligning the image and text offsets from anchor point $\mathcal{A}_i$ to target $\mathcal{T}$:

$$\mathcal{L}_{adaptive} = 1 - \cos(\Delta I_{\mathcal{A}_i \to t}, \Delta T_{\mathcal{A}_i \to \mathcal{T}}) \tag{3}$$

We empirically find that adding this adaptive loss to $\mathcal{L}_{direction}$ results in a more stable adaptation. The pseudo-code can be found in Sec. B.

## 4.2 ALIGNING PROMPT TO IMAGES

Here, we explain the details of the proposed method for learning text prompt $P_{\mathcal{A}_i}$ that describes $i^{th}$ anchor point $\mathcal{A}_i$ in text domain. Inspired by Zhou et al. (2022b); Teo et al. (2024), we define prompt $P_{\mathcal{A}_i} \in \mathbb{R}^{(M+1) \times d}$ as combination of $M$ learnable tokens $[V]_j^i \in \mathbb{R}^d$ and a label token $Y_{\mathcal{A}_i} \in \mathbb{R}^d$:

$$P_{\mathcal{A}_i} = [V]_1^i [V]_2^i \ldots [V]_M^i [Y_{\mathcal{A}_i}] \tag{4}$$

Early approaches of prompt learning directly learn the learnable tokens $[V]_j^i$ from related images (Zhou et al., 2022b;a). However, recently, ITI-GEN (Zhang et al., 2023) (proposed for fair text-to-image generation) shows that learning from the offsets is more efficient for capturing the specific attribute of interest. Inspired by this, we learn the anchor text prompt $P_{\mathcal{A}_i}$ by aligning text offset to the image offset. Here, the image offset is calculated between the current and previous anchors: $\Delta I_{\mathcal{A}_{i-1} \to \mathcal{A}_i} = E_I(G_{\mathcal{A}_i}(w)) - E_I(G_{\mathcal{A}_{i-1}}(w))$. Similarly, the text prompt offset is calculated as follows: $\Delta P_{\mathcal{A}_{i-1} \to \mathcal{A}_i} = E_T(P_{\mathcal{A}_i}) - E_T(P_{\mathcal{A}_{i-1}})$. Note that the only trainable parameter is the unknown prompt $P_{\mathcal{A}_i}$ which is learned by aligning image and prompt offsets:

$$\mathcal{L}_{align} = 1 - \cos(\Delta I_{\mathcal{A}_{i-1} \to \mathcal{A}_i}, \Delta P_{\mathcal{A}_{i-1} \to \mathcal{A}_i}) \tag{5}$$

The proposed prompt learning approach is shown in Fig. 4 (b). We remark that $P_{\mathcal{A}_i}$ is the tokenized text prompt before the CLIP text encoder, and for simplicity, we slightly abuse the notation and use $E_T(P_{\mathcal{A}_i})$ to show CLIP text embedding for anchor $\mathcal{A}_i$.

**Remark:** Given that offset misalignment is less for closer concepts, we propose to use the previous anchor point $\mathcal{A}_{i-1}$ as the source to learn the prompt for the $i^{th}$ anchor $\mathcal{A}_i$. Since consecutive anchor points are close together, the directional loss is more accurate.

---

[2]We use the same settings of $t_{thresh}$ and $t_{int}$ across all 32 experiment setups.

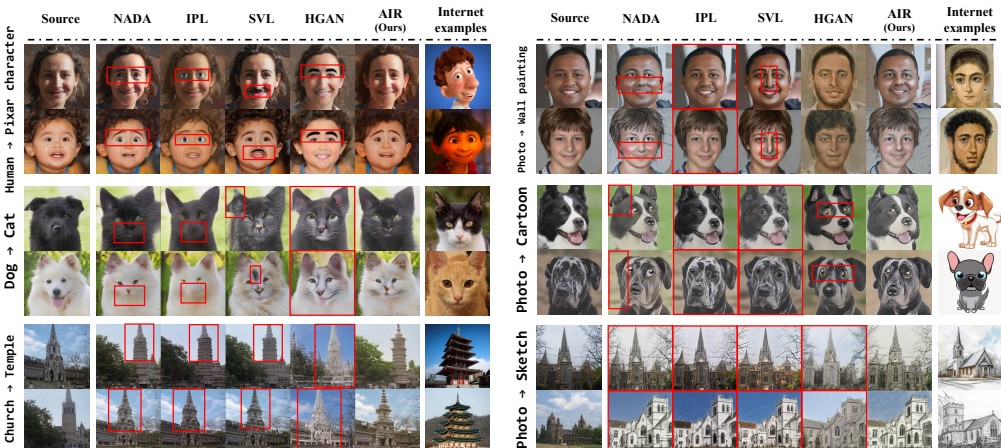

Figure 5: **Qualitative comparison** (Degraded images/regions are highlighted with red boxes). The results of NADA (Gal et al., 2022) and HGAN (Anees et al., 2024) show the adaptation often introduces undesirable changes, e.g., thick eyebrows in Human → Pixar character, missing mouth in Dog → Cat, and red cheeks in Photo → Wall painting. For IPL (Guo et al., 2023) and SVL (Jeon et al., 2023), a common issue is that the adaptations are inadequate, resulting in images that lack target features/styles, especially for adaptations that require drastic feature change, e.g., Church → Temple, Photo → Cartoon/Sketch. Our proposed method does not suffer from artifacts (as shown in target domains Pixar character, Wall painting, and Dog), and adapts better to the style of target domains, such as Temple, Cartoon and Sketch. StyleGAN2 is used as pre-trained generator. More qualitative results are shown in Fig. 1 and Sec. A.1. **(Best viewed with color and zoom in.)**

**Regularizer:** We further propose to use the interpolation between tokenized source and target descriptions as anchor label, *i.e.*, $Y_{\mathcal{A}_i} = (1 - p_i)Y_{\mathcal{S}} + p_i Y_{\mathcal{T}}$, with $p_i$ denoting the proportion of the training progress until anchor point $\mathcal{A}_i$. The label token acts like a regularizer during prompt learning (see motivation in Sec. D.2).

We empirically find that using these design choices results in better adaptation with our AIR mechanism compared to learning the prompts directly from generated images by $G_{\mathcal{A}_i}$. More details of the method are summarized in the pseudo-code in Alg. 1 and 2 in Sec. B.

## 5 EXPERIMENTS

In this section, first, we discuss the details of our experimental setup. Then, we compare our proposed AIR with SOTA methods qualitatively and quantitatively. Note that we are the first to study zero-shot adaptation of diffusion models. We also show our AIR outperforming large-scale Stable Diffusion in generating rare concepts. Finally, we conduct an ablation study on the design of the prompt learning.

### 5.1 EXPERIMENTAL SETUP

**Generative Models.** In this work, we implement zero-shot generative model adaptation for both GANs and diffusion models. The implementation details for each type of model is as follows:

- **Zero-Shot Adaptation of GANs.** We follow previous ZSGM works (Gal et al., 2022; Guo et al., 2023; Jeon et al., 2023) setups to adapt StyleGAN2-ADA (Karras et al., 2020a) pre-trained on FFHQ (Karras et al., 2019) and AFHQ-Dog (Choi et al., 2020) to various target domains.

- **Zero-Shot Adaptation of Diffusion Models.** We use Guided Diffusion (Dhariwal & Nichol, 2021) pre-trained on FFHQ and AFHQ-Dog from P2-Weighting (Choi et al., 2022) as source generator. To speedup training, we use DPM-Solver (Lu et al., 2022) for 10-step image generation. To prevent overfitting, instead of fully fine-tuning, we fine-tune with LoRA (Hu et al., 2022).

During the adaptation of both generators, we utilize the pre-trained ViT-B/32 as vision encoder for CLIP. Hyperparameter details can be found in Sec. C.4. *Notably, the only varying hyperparameter for all adaptation setups is the total number of adaptation iterations (strictly follow NADA).*

Table 1: Quantitative results of zero-shot GAN adaptation. *All methods use the same CLIP for guidance.* For FID, we report only `Baby` and `Cat`, which are the only target domains with sufficient samples for reliable FID. Note that compared with previous methods that aim to improve the synthesized sample diversity, our method (AIR) focuses on enhancing the quality of adaptation (lower CLIP Distance and FID), leading to significant gain (e.g., average CLIP distance improves spanning from 9% to 25%, and FID improves from 88.71 in HGAN to 56.20 in our AIR for distant adaptation `Dog → Cat`). The quality enhancement is consistent for all setups. Furthermore, our method is able to maintain competitive diversity (Intra-LPIPS). See qualitative comparisons in Fig. 1, 5 and Sec. A.1.

| Pre-trained Dataset | Adaptation | CLIP Distance (↓) | | | | | Intra-LPIPS (↑) | | | | | FID (↓) | | | | |
|---|---|---|---|---|---|---|---|---|---|---|---|---|---|---|---|---|
| | | NADA | IPL | SVL | HGAN | AIR | NADA | IPL | SVL | HGAN | AIR | NADA | IPL | SVL | HGAN | AIR |
| FFHQ | Human → Baby | 0.3327 | 0.3562 | 0.3838 | 0.3596 | **0.3325** | 0.4474 | 0.4518 | 0.4506 | 0.4110 | **0.4520** | 68.35 | 68.48 | 158.76 | 123.55 | **62.13** |
| | Human → Pixar | 0.2335 | 0.2343 | 0.4224 | 0.2418 | **0.2213** | **0.4759** | 0.4488 | 0.4618 | 0.4013 | 0.4717 | - | - | - | - | - |
| | Human → Werewolf | 0.3467 | 0.3200 | 0.3998 | 0.3424 | **0.2431** | 0.4301 | 0.4387 | 0.4316 | 0.4395 | **0.4410** | - | - | - | - | - |
| | Photo → Wall painting | 0.4382 | 0.4898 | 0.4952 | **0.3747** | 0.4306 | 0.4217 | 0.4320 | 0.4332 | 0.4208 | **0.4381** | - | - | - | - | - |
| | Photo → Sketch | 0.3606 | 0.3955 | 0.4092 | 0.3327 | **0.3126** | 0.4190 | 0.4292 | **0.4476** | 0.4354 | 0.4257 | - | - | - | - | - |
| | Photo → Watercolor | 0.3548 | 0.3621 | 0.3639 | 0.3865 | **0.3376** | 0.4598 | **0.4671** | 0.4612 | 0.4544 | 0.4656 | - | - | - | - | - |
| | Photo → Ukiyo-e | 0.2437 | 0.2467 | 0.3906 | 0.3286 | **0.2315** | 0.4583 | 0.4652 | 0.4406 | 0.4647 | **0.4670** | - | - | - | - | - |
| Dog | Dog → Cat | 0.1493 | 0.1530 | 0.1644 | 0.1642 | **0.1320** | 0.4439 | 0.4522 | 0.4547 | 0.4436 | **0.4628** | 70.87 | 83.29 | 65.79 | 88.71 | **56.20** |
| | Dog → Hamster | 0.1616 | 0.1457 | 0.1826 | 0.1282 | **0.1306** | 0.4196 | **0.4340** | 0.3918 | 0.3822 | 0.4213 | - | - | - | - | - |
| | Dog → Capybara | 0.1359 | 0.1446 | 0.1861 | 0.1543 | **0.1121** | 0.4312 | 0.4377 | 0.4264 | 0.4217 | **0.4401** | - | - | - | - | - |
| | Dog → Wolf | 0.1480 | 0.1519 | 0.2249 | **0.1199** | 0.1421 | 0.4305 | 0.4261 | 0.4272 | 0.4056 | **0.4349** | - | - | - | - | - |
| | Dog → the Joker | 0.3708 | 0.3827 | 0.4574 | 0.3230 | **0.3131** | 0.4155 | 0.4206 | **0.4327** | 0.4058 | 0.4173 | - | - | - | - | - |
| | Photo → Cartoon | 0.2433 | 0.2419 | 0.2543 | 0.2936 | **0.2258** | 0.4356 | 0.4413 | 0.4400 | 0.3741 | **0.4427** | - | - | - | - | - |
| | Photo → Watercolor | 0.1535 | 0.1711 | 0.1646 | 0.1611 | **0.1507** | 0.4639 | **0.4703** | 0.4622 | 0.4566 | 0.4667 | - | - | - | - | - |
| Church | Church → Skyscraper | 0.3860 | 0.3441 | 0.4209 | 0.3631 | **0.3270** | 0.4823 | 0.481 | 0.4777 | 0.4433 | **0.4839** | - | - | - | - | - |
| | Church → Temple | 0.3197 | 0.3632 | 0.3177 | 0.3229 | **0.3146** | 0.4689 | 0.4623 | 0.4839 | 0.5103 | **0.4862** | - | - | - | - | - |
| | Photo → Sketch | 0.2948 | 0.3070 | 0.3401 | 0.2957 | **0.2805** | 0.5084 | **0.5373** | 0.5356 | 0.5085 | 0.5176 | - | - | - | - | - |
| | Photo → Anime | 0.2067 | 0.2180 | 0.3503 | 0.3504 | **0.1990** | 0.5223 | **0.5392** | 0.5356 | 0.5071 | 0.5245 | - | - | - | - | - |
| | Avg. | 0.2711 | 0.2793 | 0.3293 | 0.2802 | **0.2465** | 0.4519 | 0.4575 | 0.4552 | 0.4359 | **0.4588** | 69.91 | 75.86 | 112.28 | 106.13 | **59.16** |

**Evaluation Metrics.** Following ZSGM literature, we conduct both visual inspections for qualitative evaluations and quantitative evaluations. Specifically, we evaluate image quality with FID and CLIP Distance and measure diversity using Intra-LPIPS (Ojha et al., 2021). We introduce additional metrics in Sec. A.1 to further refine quality assessment. **Additionally, a user study compares image quality and diversity across different schemes based on human feedback. (See Sec. C.5 for details.)**

## 5.2 GENERATIVE MODEL ADAPTATION

**Qualitative results.** We compare with SOTA ZSGM methods NADA (Gal et al., 2022), IPL (Guo et al., 2023), SVL (Jeon et al., 2023), and SOTA one-shot generative model adaptation method HGAN (Anees et al., 2024) (trained with both references images as it supports two-shot) as shown in Fig. 1 and 5. The results of NADA and HGAN often introduces undesirable changes in features. For IPL and SVL, the adaptations are inadequate, resulting images lack target domain feature/style. See discussion in caption for details. Our proposed method adapt correctly to target domain. Additional qualitative results of diffusion model adaptation (Sec. A.2) and GAN adaptation (Sec. A.1).

**Quantitative results.** We report FID, Intra-LPIPS, and CLIP Distance to quantify the performance of zero-shot adaptation for GAN (Tab. 1). Our method significantly outperforms SOTA in quality while maintaining competitive diversity. Our user study results in Tab. 3 further confirm the improvement of our method (details in Sec. L). Results for diffusion model (Tab. 2) show similar improvement.

**Additional Experiments.** We conduct additional experiments to demonstrate the well-behaved latent space of the pre-trained generator is preserved with our proposed approach. More specifically, we perform latent space interpolation (Sec. I) , cross-model interpolation (Sec. J) , and cross-domain image manipulation (Sec. K). We also conduct an experiment to show our method effectively learned anchor prompts (Sec. E) and effectively reduces the offset misalignment (Sec. F).

## 5.3 GENERATING RARE CONCEPTS

Notably, our proposed AIR method outperforms large-scale Stable Diffusion (SD) (Rombach et al., 2022) in generating rare concepts. As shown in Fig. 6, SD suffers from severe mode collapse, often generating nearly identical individuals. In contrast, AIR leverages source-domain diversity to enrich rare concept generation, confirming its advantage in handling challenging rare-concept scenarios. See more discussion between generative model adaptation and SD in Sec. M.

## 5.4 ABLATION STUDY

We conduct an ablation study to verify the effectiveness of our introduced prompt learning to infer text prompts for anchor points. We compare: i) $\mathcal{I} \to \mathcal{T}$: Following IPL to learn a mapper that

Table 2: Quantitative results of zero-shot diffusion model adaptation. Our AIR focuses on enhancing the quality of adaptation instead of improving synthesized sample diversity. FID is reported only for Baby and Cat, which have sufficient data for reliable evaluation. (Qualitative results in Sec. A)

| Pre-trained Dataset | Adaptation | CLIP Distance (↓) | | Intra-LPIPS (↑) | | FID (↓) | |
|---|---|---|---|---|---|---|---|
| | | NADA | AIR | NADA | AIR | NADA | AIR |
| FFHQ | Human → Baby | 0.2598 | **0.2162** | 0.5700 | **0.5779** | 65.54 | **58.05** |
| | Human → Werewolf | 0.2782 | **0.2318** | **0.5208** | 0.5195 | - | - |
| | Human → Pixar character | 0.4316 | **0.3881** | **0.4585** | 0.4549 | - | - |
| | Photo → Sketch | 0.4405 | **0.3576** | **0.4868** | 0.4860 | - | - |
| | Photo → Wall painting | 0.4791 | **0.4771** | 0.5259 | **0.5283** | - | - |
| | Photo → Watercolor | 0.3266 | **0.3234** | 0.6221 | **0.6405** | - | - |
| | Photo → Ukiyo-e | 0.2300 | **0.2141** | **0.5429** | 0.5532 | - | - |
| Dog | Dog → Cat | 0.1406 | **0.1402** | 0.5423 | **0.5445** | 85.02 | **77.61** |
| | Dog → Wolf | 0.1449 | **0.1364** | **0.4684** | 0.4726 | - | - |
| | Dog → Hamster | 0.1850 | **0.1580** | 0.4888 | **0.4961** | - | - |
| | Dog → Capybara | 0.1573 | **0.1191** | **0.4785** | 0.4596 | - | - |
| | Photo → Cartoon | 0.2544 | **0.2472** | 0.5574 | **0.5603** | - | - |
| | Photo → Watercolor | 0.1916 | **0.1848** | 0.5216 | **0.5283** | - | - |
| | Avg. | 0.2707 | **0.2456** | 0.5218 | **0.5247** | 75.28 | **67.82** |

Table 3: Results of our user study (%). Note that compared with previous methods that aim to improve diversity, our method focuses on enhancing the quality, while maintaining competitive diversity.

| Evaluation | NADA | IPL | SVL | HGAN | AIR |
|---|---|---|---|---|---|
| Quality | 24.8 | 4.2 | 3.6 | 11.7 | **55.7** |
| Diversity | 22.4 | **32.8** | 10.8 | 3.4 | 30.6 |

Table 4: Ablation study on prompt learning scheme. Visual ablation results in Sec. D.3.

| Methods | Human → Baby | | Dog → Cat | |
|---|---|---|---|---|
| | FID (↓) | Intra-LPIPS (↑) | FID (↓) | Intra-LPIPS (↑) |
| NADA | 68.35 | 0.4474 | 70.87 | 0.4439 |
| $\mathcal{I} \to \mathcal{T}$ | 98.35 | 0.4308 | 104.59 | 0.4452 |
| $\mathcal{S} \to \mathcal{A}_i$ | 64.39 | 0.4503 | 61.75 | **0.4630** |
| $\mathcal{A}_{i-1} \to \mathcal{A}_i$ | **62.13** | **0.4520** | **56.20** | 0.4628 |

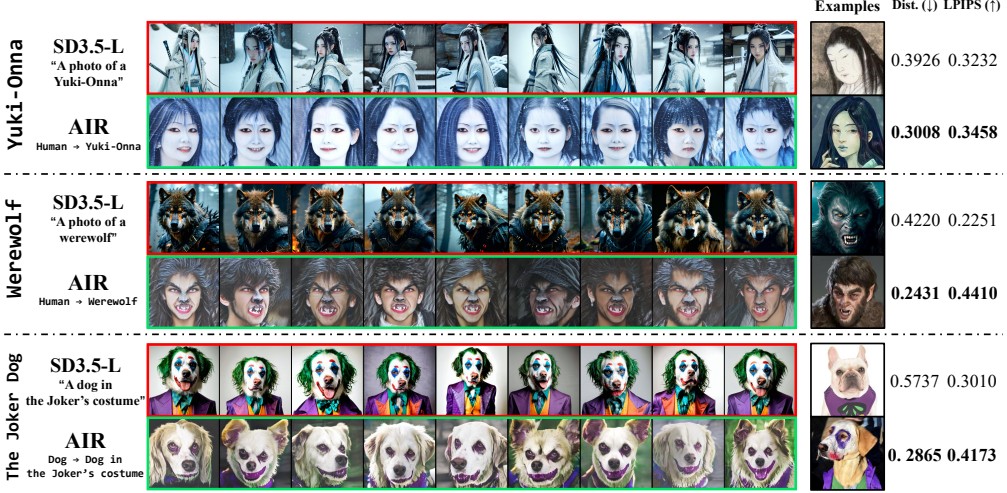

Figure 6: **Our proposed AIR outperforms SD in generating rare concepts.** We compare SoTA **SD3.5-L** with our proposed AIR. Due to the scarcity of rare concept training samples, SD generated samples have limited diversity and suffer from mode collapse: e.g., the same woman with the same hairstyle (Yuki-Onna) or nearly identical wolf/dog faces (Werewolf, Joker Dog). In contrast, ZSGM, such as our AIR, can leverage source-domain diversity to enrich generation for rare target concepts, producing variations across ages, genders, hairstyles, breeds, etc., achieving better diversity than SD in these challenging cases. **(Best viewed with color and zoom in.)**

produces prompt descriptions from each image. ii) $\mathcal{S} \to \mathcal{A}_i$: We learn the prompt by capturing the semantic difference between $\mathcal{S}$ and $\mathcal{A}$ with directional loss: $\mathcal{L}_{align}^{\mathcal{S}} = 1 - \cos(\Delta I_{\mathcal{S} \to \mathcal{A}_i}, \Delta P_{\mathcal{S} \to \mathcal{A}_i})$. iii) $\mathcal{A}_{i-1} \to \mathcal{A}_i$: Our proposed prompt learning scheme, which captures the semantic difference between consecutive anchors $\mathcal{A}_{i-1}$ and $\mathcal{A}_i$ with our proposed directional loss (Eq. 5). The results shown in Tab. 4 demonstrate our prompt learning design reduces offset misalignment compared to other schemes, therefore, leading to more accurate prompts and better zero-shot adaptation. Our visual ablation results in Sec. D.3 further confirm this observation. More ablation studies in Sec. D

## 6 CONCLUSION

All previous methods in ZSGM assume that image offset and text offset are well aligned in CLIP embedding space. In this paper, we conduct an empirical study to analyze the misalignment between image offset and text offset in CLIP space. Our analysis reveals that there is offset misalignment in CLIP space which positively correlates with concept distances. Building on this insight, we propose AIR, a new approach that iteratively samples anchor points closer to the target and reduces offset misalignment. Extentsive experimental results shows that the proposed AIR achieves SOTA performance across 32 setups. **See Supp. for limitation and societal impact.**

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

SUPPLEMENTARY

In this supplementary material, we provide additional experiments, ablation studies, and reproducibility details to support our findings. These sections are not included in the main paper due to space constraints.

Please find the following anonymous link for code and other resources: [https://anonymous.4open.science/r/AIR-15D2/](https://anonymous.4open.science/r/AIR-15D2/).

CONTENTS

## A   MORE EXPERIMENTAL RESULTS

We include a total of **32** different configurations of zero-shot adaptation in this paper. The experimental setting and evaluation metric follow Sec. 5 in the main paper.

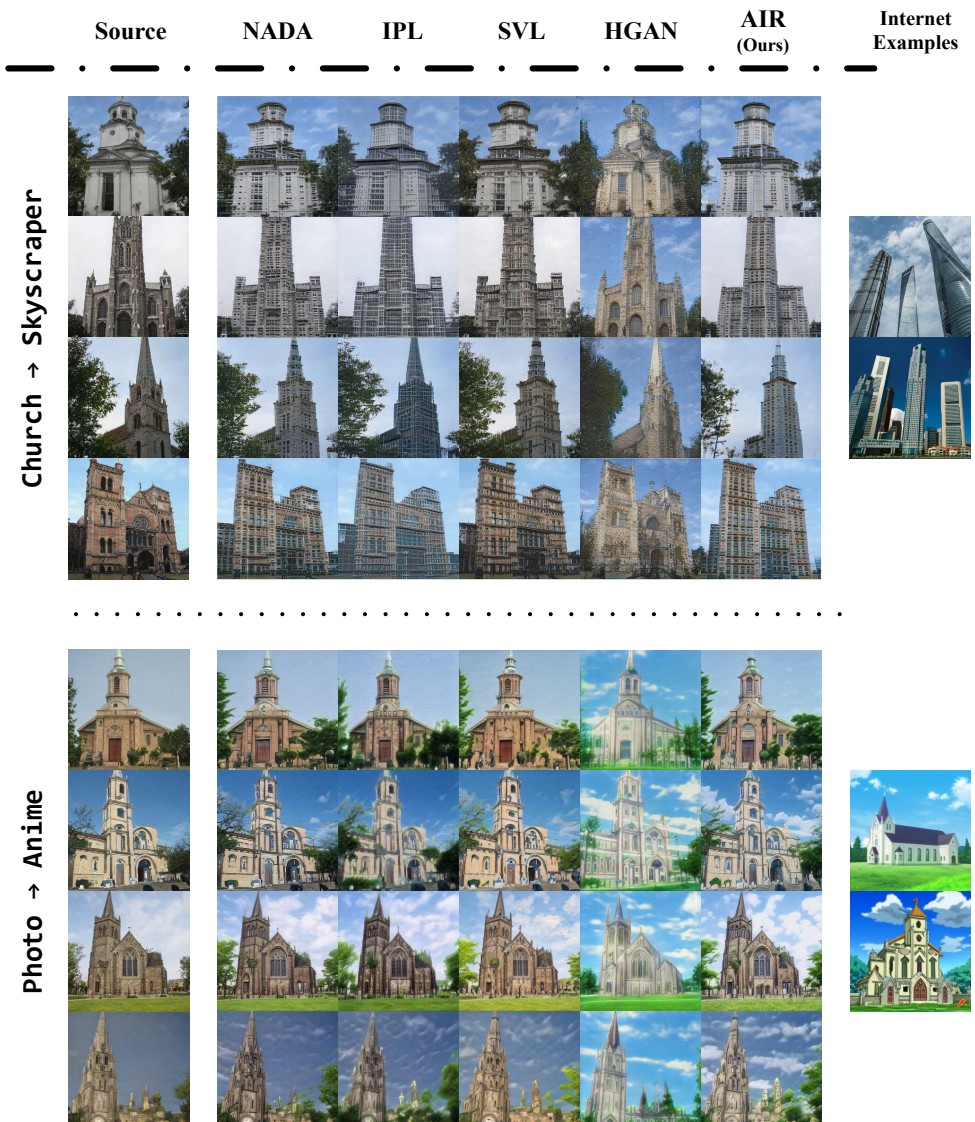

Figure 7: **Additional zero-shot adaptation results from source domain** `Church`. Here we use a StyleGAN2 generator pre-trained on the LSUN-Church (Yu et al., 2015) dataset as $G_S$ and shift this to various target domains using different zero-shot approaches. We report the qualitative results for two setups: `Church→Skyscraper` and `Photo→Anime`. We also compute CLIP Distance on 5K generated samples as quantitative results, as shown in Tab. 1, our proposed AIR approach results in less CLIP Distance meaning that the generated images are closer to the target domain. Additionally, qualitative results show that in general our proposed method adapts better to the target domain and has better quality. For example, in line 2 of `Church→Skyscraper`, NADA and IPL samples contain artifacts around windows, and SVL still has some structures related to the church like the arch in the middle of the skyscraper.

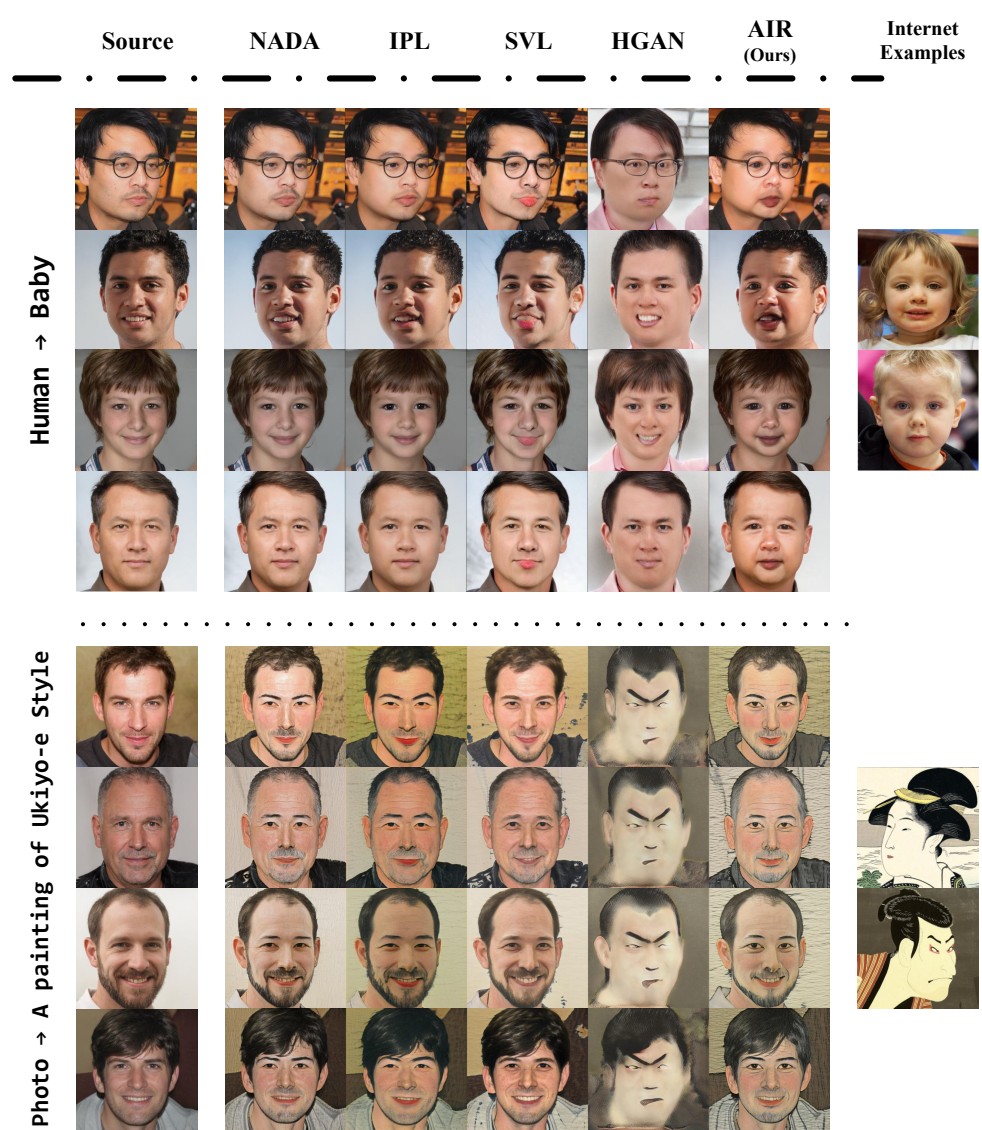

Figure 8: **Additional zero-shot adaptation results from source domain** FFHQ. Here we use a StyleGAN2 generator pre-trained on the FFHQ (Karras et al., 2019) (human faces) dataset as $G_{\mathcal{S}}$ and shift this to various target domains using different zero-shot approaches. We report the qualitative results for two setups: Photo→Baby and Photo→A Painting of Ukiyo-e Style. We also compute CLIP Distance on 5K generated samples as quantitative results, as shown in Tab. 1, our proposed AIR approach results in less CLIP Distance meaning that the generated images are closer to the target domain. Additionally, qualitative results show that in general our proposed method adapts better to the target domain and has better quality.

## A.1 ZERO-SHOT GAN ADAPTATION

In this section, we provide additional experimental results including quantitative and qualitative results for different adaptation setups using GAN as the generator and introduce more evaluation metrics.

**Qualitatives Results.** In Fig. 7 , we perform zero-shot adaptation of a StyleGAN2 pre-trained on LSUN-Church (Yu et al., 2015) to four different target domains including Skyscraper, and Anime.

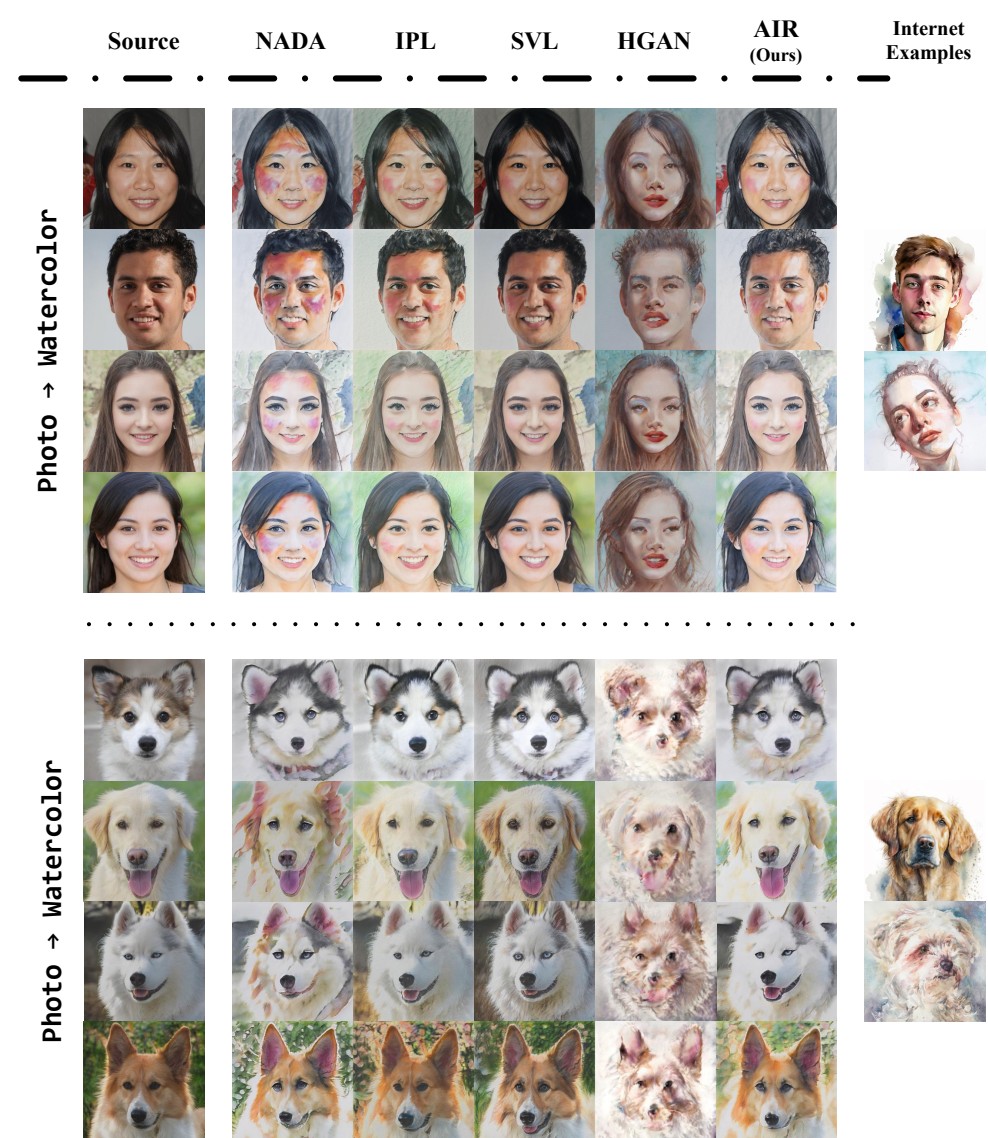

Figure 9: **Additional zero-shot adaptation results from source domain** FFHQ. Here we use a StyleGAN2 generator pre-trained on the FFHQ (Karras et al., 2019) (human faces) and AFHQ-Dog (Choi et al., 2020) (dog face) dataset as $G_S$ and shift this to various target domains using different zero-shot approaches. We report the qualitative results for the setups: Photo→Watercolor. We also compute CLIP Distance on 5K generated samples as quantitative results, as shown in Tab. 1, our proposed AIR approach results in less CLIP Distance meaning that the generated images are closer to the target domain. Additionally, qualitative results show that in general our proposed method adapts better to the target domain and has better quality.

In Fig. 8 and Fig. 9 show the qualitative results of zero-shot adaptation of a StyleGAN2 pre-trained on FFHQ (Karras et al., 2019) dataset to two different target domains including Baby, Watercolor, and A Painting of Ukiyo-e Style. Finally, Fig. 9, Fig. 10 and Fig. 11 we report the zero-shot adaptation of a StyleGAN2 pre-trained on AFHQ-Dog (Choi et al., 2020) to four different target domains including Watercolor, Hamster, Capybara, Wolf and The Joker. The results show that our approach in general adapts better to the style of the target domain and has better sample quality (please check the caption of each image for more detailed discussion).

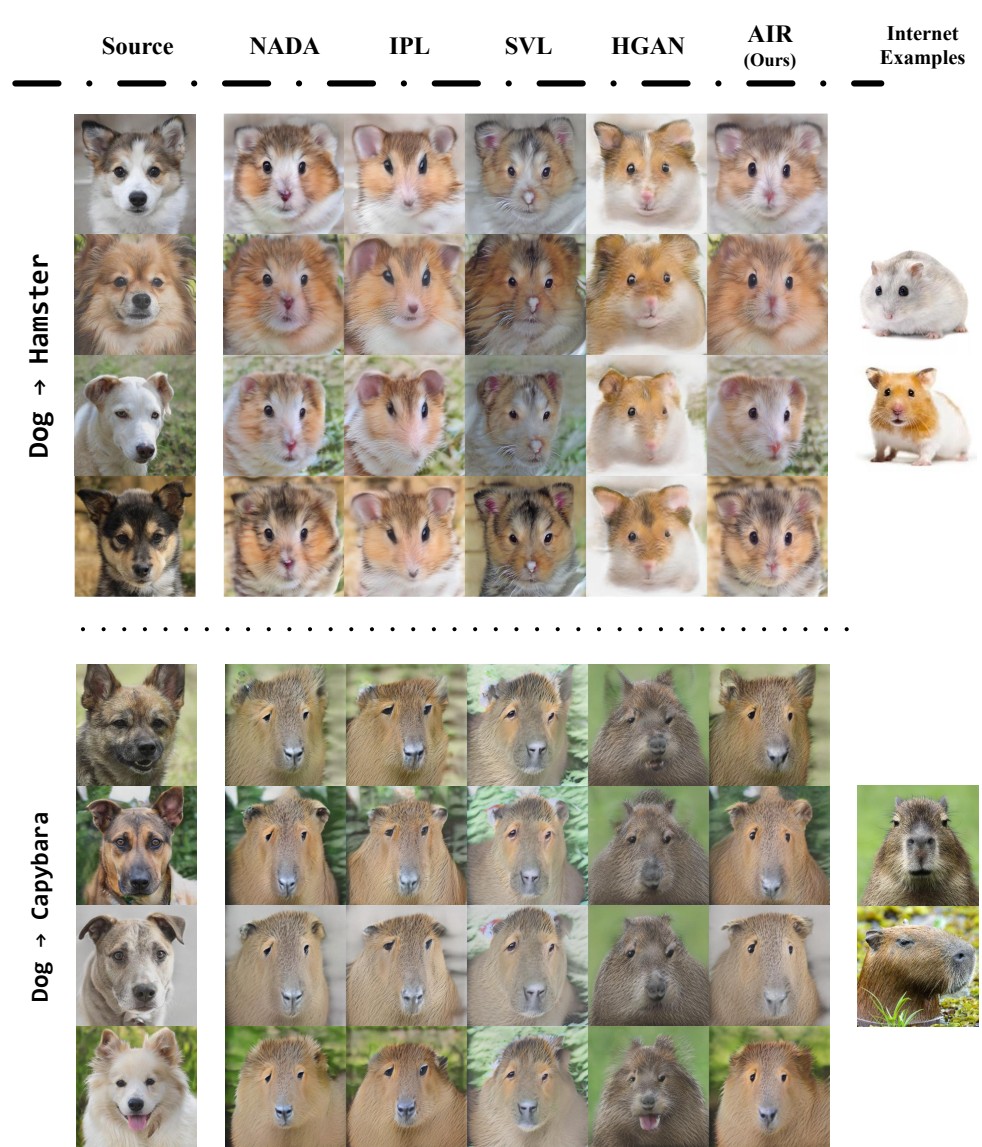

Figure 10: **Additional zero-shot adaptation results from source domain** Dog. Here we use a StyleGAN2 generator pre-trained on the AFHQ-Dog (Choi et al., 2020) dataset as $G_S$ and shift this to various target domains using different zero-shot approaches. We report the qualitative results for two setups: Dog→Hmaster and Dog→Capybara. We also compute CLIP Distance on 5K generated samples as quantitative results, as shown in Tab. 1, for both setups, our proposed AIR approach results in less CLIP Distance meaning that the generated images are closer to the target domain. Additionally, qualitative results show that in general our proposed method adapts better to the target domain and has better quality. For example, for Dog→Capybara setup, generated samples with other approaches have degradations like unsymmetrical faces or eyes.

**Quantitative Results.** Quantitative results are reported by computing the CLIP Distance between the embeddings of 5K generated images with each approach and the embedding of the text description of the target domain in CLIP space. As the results show, generated images by proposed AIR has smaller CLIP distance meaning that these images are closer to the target domain compared to images generated by other zero-shot approaches.

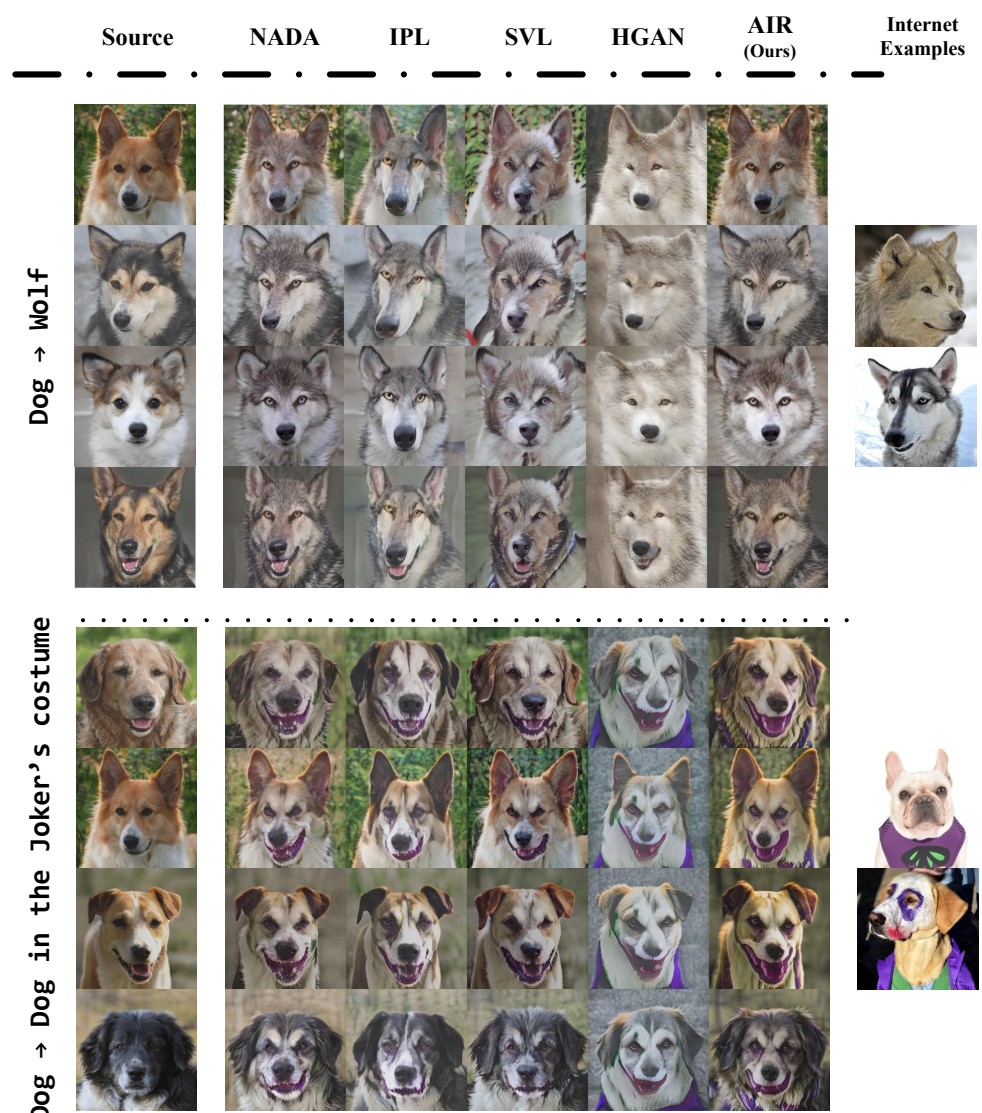

Figure 11: **Additional zero-shot adaptation results from source domain** `Dog`. Here we use a StyleGAN2 generator pre-trained on the AFHQ-Dog (Choi et al., 2020) dataset as $G_S$ and shift this to various target domains using different zero-shot approaches. We report the qualitative results for two setups: `Dog`→`Wolf` and `Dog`→`The Joker`. We also compute CLIP Distance on 5K generated samples as quantitative results, as shown in Tab. 1, for both setups, our proposed AIR approach results in less CLIP Distance meaning that the generated images are closer to the target domain. Additionally, qualitative results show that in general our proposed method adapts better to the target domain and has better quality. For example, for `Dog`→`Wolf` setup, IPL generates an unnaturally big snout and SVL has some artifacts in the generated sample. For `Dog`→`The Joker` setup, our approach attains the mouth feature with a proper style and quality.

**Additional Metrics.** We further evaluate the quality of the generated images by introducing two additional metrics SigLIP Distance (Zhai et al., 2023) and DINOv2 Distance (Oquab et al., 2024). Similar to the computation of CLIP Distance, SigLIP and DINOv2 Distance are defined as the cosine distance between the SigLIP/DINOv2 embeddings of collected and generated images. As shown in Tab. 5, the results align with those in the main paper, further support the superiority of our proposed AIR.

Table 5: Additional quantitative evaluation of zero-shot GAN adaptation, with the same setting of Tab. 1 in main paper.

| Pre-trained Dataset | Target Domain | SigLIP Distance ($\downarrow$) | | | | | DINOv2 Distance ($\downarrow$) | | | | |
|---|---|---|---|---|---|---|---|---|---|---|---|
| | | NADA | IPL | SVL | HGAN | AIR | NADA | IPL | SVL | HGAN | AIR |
| FFHQ | Baby | 0.1925 | 0.1884 | 0.3474 | 0.3080 | **0.1833** | 0.5943 | 0.5993 | 0.8026 | 0.7703 | **0.5887** |
| | Werewolf | 0.3192 | 0.3930 | 0.4831 | 0.2425 | **0.2274** | 0.8500 | 0.8923 | 0.9365 | 0.6345 | **0.6097** |
| | Pixar | 0.2803 | 0.2762 | 0.4582 | 0.2857 | **0.2630** | 0.6935 | **0.6690** | 0.7848 | 0.7362 | 0.6785 |
| | Sketch | 0.2897 | 0.3173 | 0.3598 | 0.2939 | **0.2837** | 0.4682 | 0.5918 | 0.6291 | 0.4703 | **0.4420** |
| | Wall painting | 0.4205 | 0.4277 | 0.4489 | **0.4052** | 0.4103 | 0.7256 | 0.7267 | 0.7836 | **0.6761** | 0.7004 |
| AFHQ-Dog | Cat | 0.1395 | 0.2287 | 0.1819 | 0.1762 | **0.1297** | 0.8553 | 0.8737 | 0.8842 | 0.8623 | **0.8338** |
| | Cartoon | 0.2580 | 0.2618 | 0.3140 | 0.3101 | **0.2518** | 0.7899 | 0.8174 | 0.9078 | 0.8513 | **0.7644** |
| | Watercolor | 0.1934 | 0.1980 | 0.2569 | 0.1916 | **0.1819** | 0.8059 | 0.8328 | 0.8330 | **0.7721** | 0.7943 |

---

**Algorithm 1:** Zero-Shot Learning using Adaptation with Iterative Refinement (AIR)

---

**Require:** Pre-trained generator $G_\mathcal{S}$, textual descriptions $T_\mathcal{S}$ and $T_\mathcal{T}$, $t_{adapt}$, $t_{thresh}$, $t_{int}$, learning rate $\eta$, CLIP image and text encoder $E_I$ and $E_T$

**Output:** Trained generator $G_t$ to produce high-quality target domain images

1 Initialize $G_t$ by weights of $G_\mathcal{S}$ and freeze weights of $G_\mathcal{S}$, $i = 0$, $\mathcal{L}_{adaptive} = 0$

2 $\Delta T_{\mathcal{S} \to \mathcal{T}} = E_T(T_\mathcal{T}) - E_T(T_\mathcal{S})$

3 **for** $t = 0$; $t{+}{+}$; $t < t_{adapt}$ **do**

4    $\Delta I_{\mathcal{S} \to t} = E_I(G_t(w)) - E_I(G_\mathcal{S}(w))$

5    $\mathcal{L}_{direction} = 1 - \cos(\Delta I_{\mathcal{S} \to t}, \Delta T_{\mathcal{S} \to \mathcal{T}})$

6    **if** $t \% t_{int} = 0$ **then**

7      $i{+}{+}$

8      $G_{\mathcal{A}_i} = G_t$

9      $P_{\mathcal{A}_i} = $ Prompt-Learning $(G_{\mathcal{A}_i}, G_{\mathcal{A}_{i-1}}, P_{\mathcal{A}_{i-1}})$ /* refer to Algorithm 2 for details */

10    **end**

11    **if** $t > t_{thresh}$ **then**

12      $\Delta I_{\mathcal{A}_i \to t} = E_I(G_t(w)) - E_I(G_{\mathcal{A}_i}(w))$ /* if $G_t = G_{\mathcal{A}_i}$, add perturbation to $G_t(w)$ */

13      $\Delta T_{\mathcal{A}_i \to \mathcal{T}} = E_T(T_\mathcal{T}) - E_T(P_{\mathcal{A}_i})$

14      $\mathcal{L}_{adaptive} = 1 - \cos(\Delta I_{\mathcal{A}_i \to t}, \Delta T_{\mathcal{A}_i \to \mathcal{T}})$

15    **end**

16    $\mathcal{L} = \mathcal{L}_{direction} + \mathcal{L}_{adaptive}$

17    Update $G_t \leftarrow G_t - \eta \nabla_{G_t} \mathcal{L}$

18 **end**

---

### A.2 ZERO-SHOT DIFFUSION MODEL ADAPTATION

In this section, we provide more qualitative and quantitative results of zero-shot diffusion model adaptation.

**Qualitative Results.** Here, we report the qualitative results of zero-shot diffusion model adaptation for the same configuration used in Tab. 2 (main paper). More specifically, we use the pre-trained Guided Diffusion model (Dhariwal & Nichol, 2021) on two different source domains FFHQ (Karras et al., 2019) (Fig. 12) and AFHQ-Dog (Choi et al., 2020) (Fig. 13) and shift these pre-trained models to different target domains using only text descriptions for both NADA and our proposed AIR approaches. As illustrated in Fig. 12 and 13, the generated images with NADA suffer from degradation in the form of artifacts compared to our proposed AIR approach.

## B ALGORITHM

We provide the pseudo-code of the proposed method in this section. Specifically, we show zero-shot generative model using Adaptation with Iterative Refinement (AIR) in Alg. 1, and our proposed prompt learning scheme in Alg. 2.

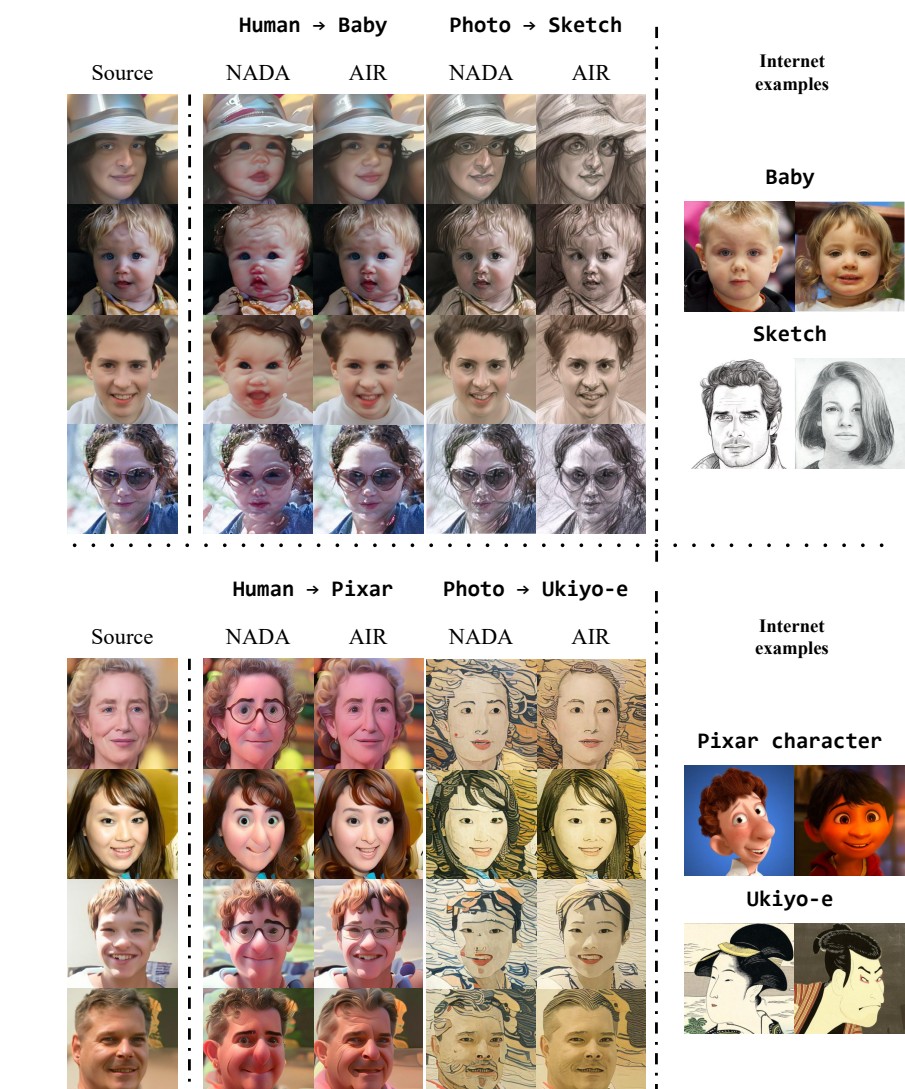

Figure 12: **Additional zero-shot adaptation results.** We use a pre-trained Guided Diffusion model (Dhariwal & Nichol, 2021) on FFHQ dataset (Karras et al., 2019) as pre-trained generator $G_\mathcal{S}$ and perform zero-shot adaptation in four different setups: Human→ Baby, Photo→ Sketch, Human→ Pixar Character, and Photo→ A painting in Ukiyo-e style using both NADA and our proposed AIR approach. Quantitative results measured by CLIP distance in 2 shows that the generated images by our approach are closer to the target domain. In addition, qualitative results show that NADA suffers from degradation.

## C    DETAILED EXPERIMENTAL SETTING

### C.1    DETAILS OF EMPIRICAL ANALYSIS

For datasets with a single class label for each image, such as ImageNet, Caltech-101, and CIFAR-100, we use the original images from the dataset. For datasets with multiple objects in an image, such as OpenImages, MS COCO, and Visual Genome, to better align with the setting in NADA, we extract the objects using bounding boxes and classify them into their labeled classes.

For a certain concept $\alpha$, we use the images of the class as $I_\alpha$. For text description $T_\alpha$, we use the corresponding class label with INt, e.g., "a photo of a [cat]" when $\alpha = $ cat.

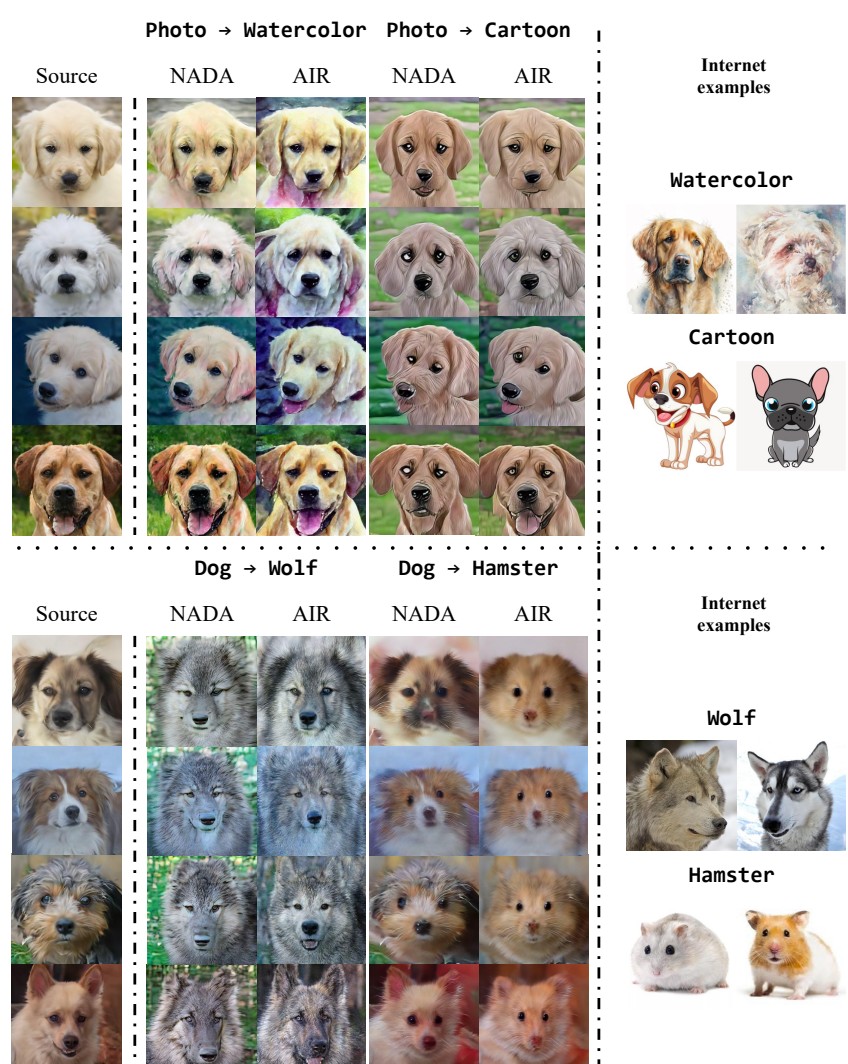

Figure 13: **Additional zero-shot adaptation results.** We use a pre-trained Guided Diffusion model (Dhariwal & Nichol, 2021) on AFHQ-Dog dataset (Choi et al., 2020) as pre-trained generator $G_S$ and perform zero-shot adaptation in two different setups: Photo→ Watercolor, Photo→ Cartoon, Dog→ Wolf, and Dog→ Hamster using both NADA and our proposed AIR approach. Quantitative results measured by CLIP distance in 2 shows that the generated images by our approach are closer to the target domain. Similarly, qualitative results show that our proposed AIR approach has better performance compared to NADA.

## C.2 Details of Impact of Offset Misalignment

We randomly sample prompt template from INt, and perform zero-shot adaptation with NADA as shown in Fig. 3 in main paper. We list the details of the sampled prompts and their offset misalignment $\mathcal{M}$ as well as the adaptation quality (measured by FID) in Tab. 6.

## C.3 Hyperparameters of Impact of Offset Misalignment

For the hyperparameter choices in Sec. 3.2 , we strictly follow the settings in NADA except that only the ViT-B/32 is used as vision encoder. The details of hyperparameters are shown in Tab. 7.

---

**Algorithm 2:** Proposed Prompt Learning

---

**Require:** Current and previous anchor generators $G_{\mathcal{A}_i}$ and $G_{\mathcal{A}_{i-1}}$, learned text prompt for
        previous anchor $P_{\mathcal{A}_{i-1}}$, learning rate $\mu$, CLIP image and text encoder $E_I$ and $E_T$

**Output:** Prompt vector $P_{\mathcal{A}_i}$ to represent current anchor.

1   $\Delta I_{\mathcal{A}_{i-1} \to \mathcal{A}_i} = E_I(G_{\mathcal{A}_i}(w)) - E_I(G_{\mathcal{A}_{i-1}}(w))$

2   **for** $k = 0;\ k{+}{+};\ k < k_{iter}$ **do**

3      $\Delta P_{\mathcal{A}_{i-1} \to \mathcal{A}_i} = E_T(P_{\mathcal{A}_i}) - E_T(P_{\mathcal{A}_{i-1}})$.

4      $\mathcal{L}_{align} = 1 - \cos(\Delta I_{\mathcal{A}_{i-1} \to \mathcal{A}_i}, \Delta P_{\mathcal{A}_{i-1} \to \mathcal{A}_i})$

5      Update $P_{\mathcal{A}_i} \leftarrow P_{\mathcal{A}_i} - \mu \nabla_{P_{\mathcal{A}_i}} \mathcal{L}_{align}$

6   **end**

---

Table 6: Prompt templates used in Sec. 3.2.

| Prompts | Human→Baby Offset Misalignment | Human→Baby FID | Dog→Cat Offset Misalignment | Dog→Cat FID |
|---|---|---|---|---|
| A bad photo of a { }. | 0.6971 | 62.76 | 0.3545 | 69.47 |
| A sculpture of a { }. | 0.7895 | 68.08 | 0.4713 | 101.49 |
| A photo of the hard to see { }. | 0.7989 | 76.36 | 0.4219 | 75.24 |
| A low resolution photo of the { }. | 0.7729 | 83.18 | 0.3942 | 76.06 |
| A rendering of a { }. | 0.7577 | 73.56 | 0.4028 | 111.74 |
| Graffiti of a { }. | 0.7715 | 92.34 | 0.5332 | 83.03 |
| A bad photo of the { }. | 0.7202 | 66.58 | 0.3774 | 66.58 |
| A cropped photo of the { }. | 0.8215 | 89.66 | 0.4512 | 132.33 |
| A tattoo of a { }. | 0.8060 | 108.78 | 0.5490 | 119.40 |
| The embroidered { }. | 0.8185 | 104.13 | 0.5514 | 109.27 |
| A photo of a hard to see { }. | 0.7680 | 74.58 | 0.4066 | 79.07 |
| A bright photo of a { }. | 0.7315 | 69.54 | 0.4305 | 77.50 |
| A dark photo of the { }. | 0.7758 | 83.50 | 0.4592 | 114.12 |
| A drawing of a { }. | 0.7765 | 89.28 | 0.4304 | 123.84 |
| A photo of my { }. | 0.6949 | 58.39 | 0.3566 | 77.76 |
| The plastic { }. | 0.7812 | 119.73 | 0.5092 | 113.99 |
| A photo of the cool { }. | 0.8094 | 103.78 | 0.4496 | 93.12 |
| A close-up photo of a { }. | 0.7213 | 69.61 | 0.4370 | 72.75 |
| A black and white photo of the { }. | 0.7463 | 64.99 | 0.5288 | 140.25 |
| A painting of the { }. | 0.8152 | 121.74 | 0.4862 | 150.15 |
| A painting of a { }. | 0.7576 | 87.01 | 0.4513 | 89.32 |
| A pixelated photo of the { }. | 0.7154 | 62.85 | 0.5168 | 105.32 |
| A sculpture of the { }. | 0.7794 | 82.22 | 0.5086 | 115.97 |
| A bright photo of the { }. | 0.8029 | 114.31 | 0.4203 | 83.28 |
| A cropped photo of a { }. | 0.7493 | 86.87 | 0.3929 | 93.22 |
| A plastic { }. | 0.7420 | 75.65 | 0.5247 | 127.82 |
| A photo of the dirty { }. | 0.8276 | 96.47 | 0.5004 | 85.62 |
| A jpeg corrupted photo of a { }. | 0.7972 | 92.56 | 0.5872 | 88.73 |

## C.4   HYPERPARAMETERS OF ZERO-SHOT ADAPTATION

In Alg. 1, for both GAN and diffusion model adaptation the batch size is set to 2. Adaptation iteration $t_{adapt}$ is set to 200 for in-domain changes like Human→Baby, 300 for texture-based changes such as Photo→Sketch, and 2,000 for animal changes like Dog→Cat. We set $t_{thresh} = 50\% t_{adapt}$ to ensure there are some target domain concept encoded in $G_t$, and $t_{int} = 10\% t_{adapt}$ to facilitate a stable and efficient training.

Table 7: Hyperparameters choices of NADA in Sec. 3.2.

| Source | Target | Prompt template | Iterations | Adaptive k |
|--------|--------|-----------------|------------|------------|
| Human | Baby | INt | 300 | 18 |
| Dog | Cat | INt | 2000 | 3 |

In Alg. 2, we generate 1,000 pairs of source and anchor images with the same batch of $w$ for each update. The number of prompt vectors $m$ is set to 4, and is initialized by "A photo of a". Each of the prompt learning sessions requires $k_{iter} = 200$ iterations.

For all experiments, we use an ADAM optimizer with a learning rate of 0.002 for both Alg.1 and 2. We conduct all the experiments on a single NVIDIA RTX 6000 Ada GPU. The training time is comparable to NADA as prompt learning in Alg. 2 only requires ∼20 seconds in our environment.

It is important to note that the only varying hyperparameter for all 26 setups is the number of adaptation iterations (same as NADA), and our results show this generalizes well across scenarios.

### C.5 EVALUATION DETAILS

A well-trained image generator is defined by its ability to produce high-quality and diverse images from target distribution. We follow existing zero-shot works in evaluation setup when applicable, and further improve on them. Specifically, following previous works (Gal et al., 2022; Guo et al., 2023; Jeon et al., 2023), we have conducted comparisons on both public datasets and images collected from the internet. Our evaluations include both visual inspections for qualitative evaluations and quantitative evaluations using the following metrics:

- **FID.** For target domains with large and publicly available datasets, we follow previous work (Jeon et al., 2023) to use FFHQ-Baby (Ojha et al., 2021) (for target domain Baby), and AFHQ-Cat (Choi et al., 2020) (for target domain Cat) as target distribution. Then, we generate 5000 samples for each target domain (Zhao et al., 2022a; 2023), and use FID to evaluate the generated images' quality and diversity.

- **CLIP Distance.** The public data is scarce for other target domains, *e.g.,* Pixar. For these domains, we follow IPL's idea (Guo et al., 2023) to collect internet images as reference. However, since IPL did not make the collected images publicly available, we had to repeat the same practice and collect the images. Then, we use the CLIP Distance (Gal et al., 2023) which is defined as the cosine distance between the clip embeddings of the collected images and the generated images to measure the similarity of the generated images to the target domain.

- **Intra-LPIPS.** To measure the diversity of the generated images, we use Intra-LPIPS metric (Ojha et al., 2021) which first assigns generated images to one of $K$ clusters, then averages pair-wise distance within the cluster members and reports the average value over $K$ clusters. In zero-shot setup, since there are no training images, we follow (Gal et al., 2022; Jeon et al., 2023) to cluster around generated images using $K$-Medoids (Kaufman & Rousseeuw, 2009), with $K = 10$.

- **User Study.** We also conducted a user study to compare the quality and the diversity of the generated images with different schemes based on human feedback. See more details in Sec. L.

We remark that similarly NADA reports Intra-LPIPS on AFHQ-Cat, and SVL reports both FID and Intra-LPIPS on AFHQ-Cat. In addition, we believe the included visual results can help in transparency and reflecting the superior performance of our proposed method in terms of adaptation quality.

## D ADDITIONAL ABLATION STUDIES

### D.1 ABLATION ON HYPERPARAMETERS SELECTION

We conduct an ablation study to determine the optimal hyperparameters. Specifically, Tab. 8 shows the ablation results for the adaptation interval $t_{int}$ to update anchor. Tab. 9 shows the ablation results for the starting iteration $t_{thresh}$ of applying AIR. A large $t_{int}$ which results in fewer updates of the anchor

point, generally leads to a degradation in performance due to less precise adaptation. Conversely, a small $t_{int}$, while more computationally expensive, does not yield significant improvement. Thus, we set $t_{int} = 10\%$ to balance the computation cost and adaptation precision. Similarly, neither excessively large nor small values of $t_{thresh}$ provide optimal adaptation performance. As shown in Fig. 14 (b) and (c), the visual ablation results align with this conclusion. Hence, we empirically select $t_{thresh} = 50\%$. It is important to note that the only varying hyperparameter for all 26 setups is the number of adaptation iterations (same as NADA), and our results show this generalizes well across scenarios.

Table 8: Ablation study on adaptation interval $t_{int}$ to update anchor.

| % of $t_{adapt}$ | $t_{int}$ | | | |
| | Human $\rightarrow$ Baby | | Dog $\rightarrow$ Cat | |
| | FID ($\downarrow$) | Intra-LPIPS ($\uparrow$) | FID ($\downarrow$) | Intra-LPIPS ($\uparrow$) |
|---|---|---|---|---|
| 5% | 59.45 | 0.4512 | 59.97 | 0.4560 |
| 10% | 62.13 | **0.4520** | **56.20** | **0.4628** |
| 15% | 58.87 | 0.4515 | 61.92 | 0.4635 |
| 20% | 64.54 | 0.4496 | 65.49 | 0.4537 |
| 25% | **56.69** | 0.4511 | 67.49 | 0.4374 |
| 30% | 76.39 | 0.4506 | 77.23 | 0.4513 |

Table 9: Ablation study on starting iteration $t_{thresh}$ of applying AIR.

| % of $t_{adapt}$ | $t_{thresh}$ | | | |
| | Human $\rightarrow$ Baby | | Dog $\rightarrow$ Cat | |
| | FID ($\downarrow$) | Intra-LPIPS ($\uparrow$) | FID ($\downarrow$) | Intra-LPIPS ($\uparrow$) |
|---|---|---|---|---|
| 0% | 63.56 | 0.4324 | 83.33 | **0.4815** |
| 12.5% | 75.92 | 0.4259 | 78.21 | 0.4642 |
| 25.0% | 72.17 | 0.4386 | 73.65 | 0.4516 |
| 37.5% | 64.14 | **0.4558** | 59.96 | 0.4496 |
| 50.0% | **62.13** | 0.4520 | **56.20** | 0.4628 |
| 67.5% | 68.25 | 0.4542 | 56.46 | 0.4344 |

### D.2 ABLATION ON ANCHOR LABEL INITIALIZATION

Our prompt initialization in Sec. 4.2 is inspired by the standard prompt learning in VLM (Zhou et al., 2022b;a), which initializes label token $Y$ with class label of the image to serve as prior. However, in our setting, the anchor domain encodes both source and target concepts, so it cannot be described with natural language. Therefore, we leverage the continuous and semantically rich embedding space of the text encoder to initialize anchor label $Y_{\mathcal{A}_i}$ via **interpolation** between tokenized source and target descriptions. We conduct an ablation study on the initialization of $Y_{\mathcal{A}_i}$. Results shown in Tab. 10 indicate the effectiveness of our idea by obtaining the best FID and Intra-LPIPS.

### D.3 VISUAL ABLATION STUDIES

We perform visual ablation studies on prompt learning design and hyperparameters selection with the same experiments setting of Sec. 5.4 and D.1. The results in Fig. 14 align consistently with the quantitative findings.

## E VALIDATE OUR LEARNED ANCHOR PROMPTS

To validate our prompt learning, we visualize the learned prompts and generated anchor domain images in CLIP space. As shown in Fig. 15, the prompts accurately represent the anchors (3 of 5 anchors shown for clarity).

Table 10: Ablation study on initialization of $Y_{\mathcal{A}_i}$, using: a) Target domain label; b) Source domain label for the first half of adaptation, then target domain label; c) Interpolation as in our AIR.

| Init. | Human → Baby | | Dog → Cat | |
|---|---|---|---|---|
| | FID ($\downarrow$) | Intra-LPIPS ($\uparrow$) | FID ($\downarrow$) | Intra-LPIPS ($\uparrow$) |
| a) | 67.53 | 0.4513 | 65.83 | 0.4373 |
| b) | 63.34 | 0.4512 | 56.44 | 0.4466 |
| c) | **62.13** | **0.4520** | **56.20** | **0.4628** |

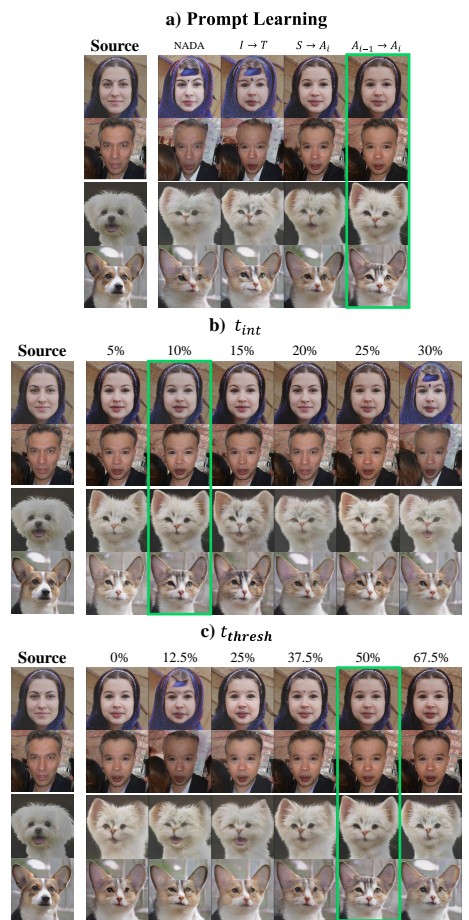

Figure 14: Visual ablation study of: a) Design choice of prompt learning; b) Adaptation interval $t_{int}$ to update anchor; c) Starting iteration $t_{thresh}$ of applying AIR.

## F  OFFSET MISALIGNMENT ALLEVIATION

We demonstrate AIR alleviates the offset misalignment, i.e., our refined direction aligns more with ground truth in Tab. 11. The ground truth is computed by $\Delta I_{\mathcal{S} \to \mathcal{T}} = E_I(\overline{I_\mathcal{T}}) - E_I(\overline{G_\mathcal{S}(w)})$ (for AIR, the ground truth is $\Delta I_{\mathcal{A}_i \to \mathcal{T}}$ of the last $\mathcal{A}_i$), where $I_\mathcal{T}$ are real images.

## G  OFFSET MISALIGNMENT IN OTHER MULTIMODAL REPRESENTATION SPACES

In this section, we present an additional empirical analysis of offset misalignment for other contrastive learning-based Multimodal Representation spaces. Following the experimental setup in Sec. 3.1, but

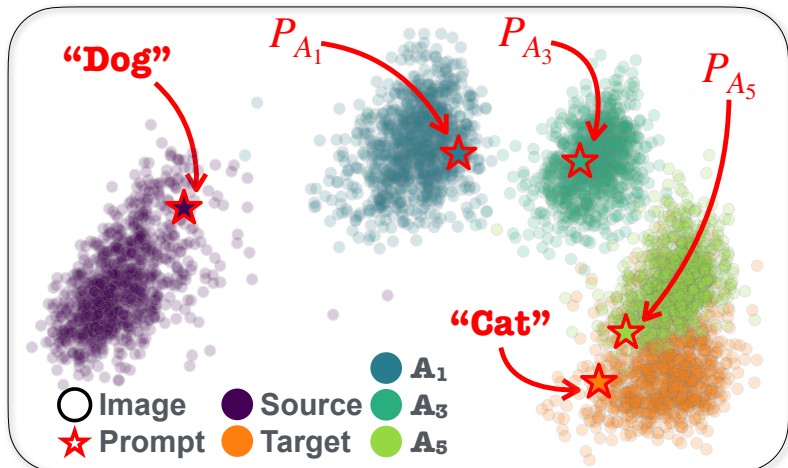

Figure 15: PCA visualization for Dog → Cat. For each anchor point $\mathcal{A}_i$, our learned prompt $P_{\mathcal{A}_i}$ lies within the distribution of 1000 generated images by the generator $G_{\mathcal{A}_i}$ for that anchor point.

Table 11: Offset misalignment between adaptation directions and the ground truth. Note that IPL and SVL have multiple directions.

| Adaptation | NADA | IPL | SVL | AIR |
|---|---|---|---|---|
| Human → Baby | 0.67 | $0.69_{\pm 0.09}$ | $0.92_{\pm 0.03}$ | **0.49** |
| Dog → Cat | 0.54 | $0.65_{\pm 0.06}$ | $0.59_{\pm 0.11}$ | **0.25** |

replacing the CLIP ViT-Base/32 vision encoder with CLIP ConvNext-L, CLIP RN50x64, and SigLIP ViT-L/16-256, we plot the offset misalignment against concept distance for six public datasets in Fig. 16, Fig. 17, and Fig. 18. Our results demonstrate consistent and meaningful positive correlations between offset misalignment and concept distance across different CLIP-like spaces.

## H  CONCEPT SHIFTS DURING ADAPTATION

The intuition of our proposed method is that after limited iterations of adaptation using directional loss, the encoded concept in the adapted generator is already closer to the target domain than the encoded concept in source generator. In this section, we design an experiment to demonstrate that the adapted generator already encodes some knowledge related to the target domain. Specifically, following zero-shot generative model domain adaptation setup (Gal et al., 2022), we perform adaptation on Human→Baby with StyleGAN2-ADA pretrained on FFHQ (Karras et al., 2019). We report FID score throughout the adaptation process to measure the knowledge related to target domain encoded in the adapted generator. Our results in Fig. 19 support our statement. Additionally, we present qualitative results using the same latent code to further support our findings.

## I  LATENT SPACE INTERPOLATION

Building on prior research, we demonstrate that the target domain generators refined through our method retain a smooth latent space property. As illustrated in Fig. 20, each row features a series of images from the same target domain. The left-most and right-most images in each row, labeled as $G_t(w_1)$ and $G_t(w_2)$ respectively, are generated using distinct latent codes $w_1$ and $w_2$. Latent space interpolation between these codes produces an image $G_t((1 - \gamma)w_1 + \gamma w_2)$, where $\alpha$ varies from 0 to 1. The visual results show that our method has good robustness and generalization ability. The various target domain spaces obtained by our method are consistently smooth.

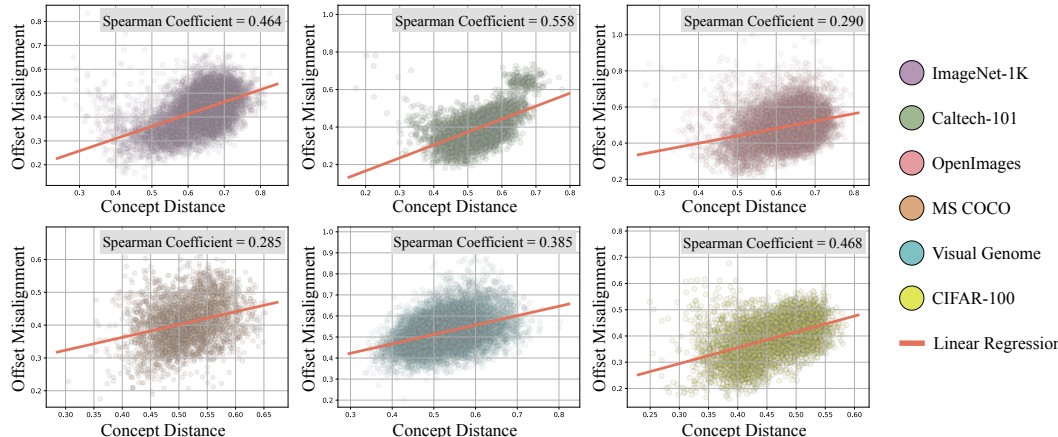

Figure 16: Empirical analysis of offset misalignment in CLIP ConvNext-L space. Experiment setup is the same as Sec. 3.1 except that CLIP ConvNext-L is used as the vision encoder. Our results show that the meaningful correlation (measured by Spearman's coefficient (Zar, 2005)) between offset misalignment and concept distance consistently exists in both ConvNext-based and ViT-based CLIP vision encoders.

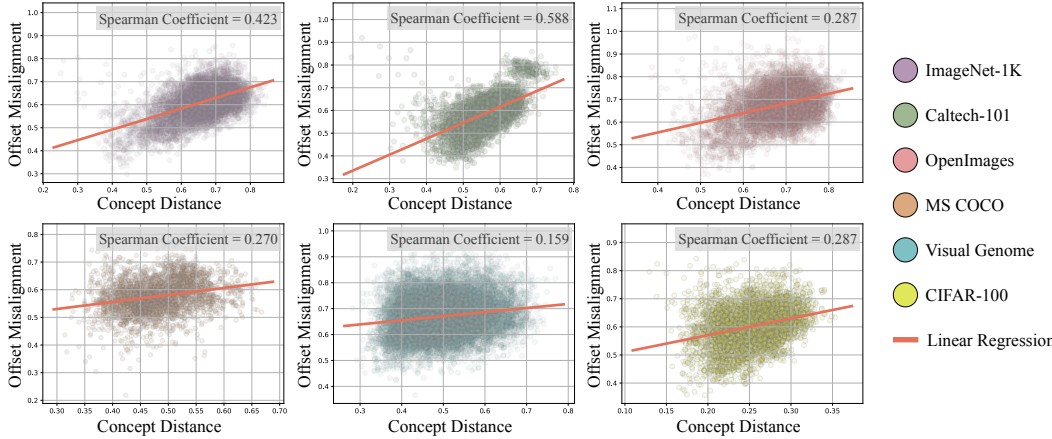

Figure 17: Empirical analysis of offset misalignment in CLIP RN50x64 space. Experiment setup is the same as Sec. 3.1 except that CLIP RN50x64 is used as the vision encoder. Our results show that the meaningful correlation (measured by Spearman's coefficient (Zar, 2005)) between offset misalignment and concept distance consistently exists in both CNN-based and ViT-based CLIP vision encoders.

## J   CROSS-MODEL INTERPOLATION

In addition to demonstrating latent space interpolation, we also explore the model's weight smoothness across various domains. Specifically, we perform linear interpolation in the weight space between $G(\cdot, \theta_s)$ and $G(\cdot, \theta_{t_1})$, or between $G(\cdot, \theta_{t_1})$ and $G(\cdot, \theta_{t_2})$. Here, $G(\cdot, \theta_s)$ represents the source domain generator, while $G(\cdot, \theta_{t_1})$ and $G(\cdot, \theta_{t_2})$ are generators adapted to two different target domains. Given a latent code $w$, we produce images via an interpolated model, $G(w, (1 - \gamma)\theta_1 + \gamma\theta_2)$, where $\gamma$ ranges from 0 to 1. As illustrated in Fig. 21, our approach effectively supports smooth cross-model interpolation, whether transitioning from a source to a target domain or between different target domains.

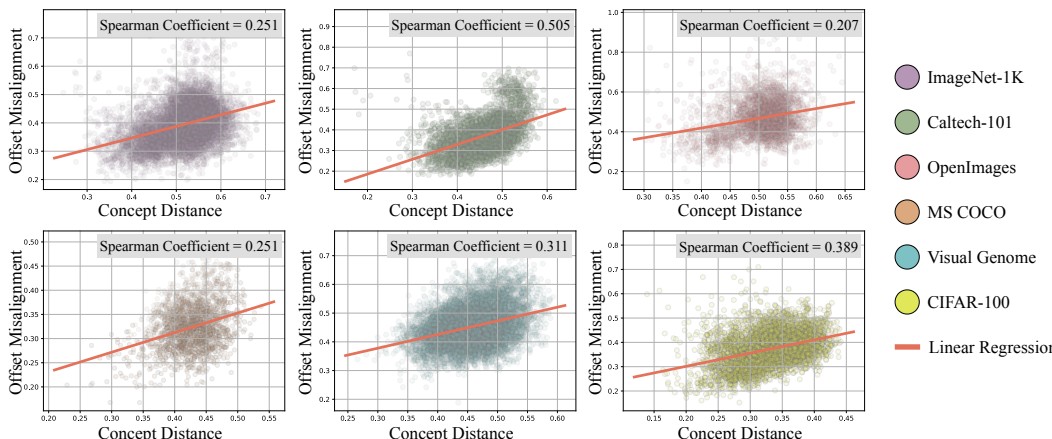

Figure 18: Empirical analysis of offset misalignment in SigLIP ViT-L/16-256 space. Experiment setup is the same as Sec. 3.1 except that SigLIP ViT-L/16-256 is used as the vision encoder. Our results show that the meaningful correlation (measured by Spearman's coefficient (Zar, 2005)) between offset misalignment and concept distance consistently exists in various contrastive learning-based multimodal vision encoders.

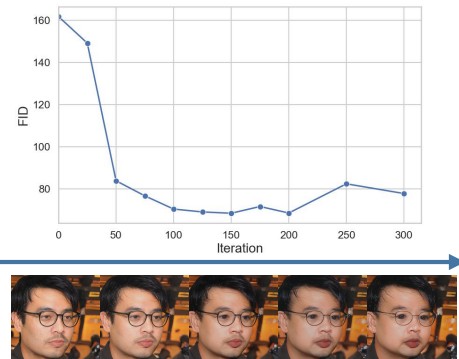

Figure 19: Concept shifts during adaptation.

## K  IMAGE MANIPULATION

To further demonstrate the effectiveness of our proposed method, we also conduct experiments on text-to-image manipulation. It first inverts a image to the latent code by a pre-trained inversion model and then feeds it to the trained target domain generator to get the translated target domain image.

We experiment on both GAN and diffusion model. We use Restyle (Alaluf et al., 2021) with e4e encoder (Tov et al., 2021) to invert a real image into the latent space $w$ for StyleGANs. For the diffusion model, we follow the setting of DiffusionCLIP (Kim et al., 2022) to diffuse a real image and fintune the model to generate an image with target domain features using the diffused image.

### K.1  GAN-BASED IMAGE MANIPULATION

For GAN-based generators, we perform the experiment by utilizing the inversion model Restyle (Alaluf et al., 2021) with e4e encoder (Tov et al., 2021). As illustrated in Fig 22, our method qualitatively exhibits a higher fidelity of target domain features compared to previous methods. Quantitatively, our approach more closely aligns with the reference target images in CLIP space, indicating a greater semantic similarity.

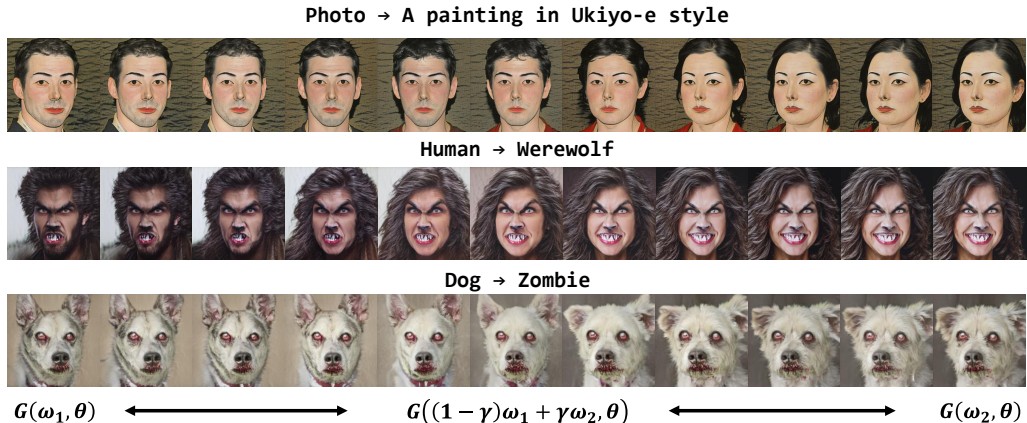

$$G(\omega_1, \theta) \longleftrightarrow G((1-\gamma)\omega_1 + \gamma\omega_2, \theta) \longleftrightarrow G(\omega_2, \theta)$$

Figure 20: Latent space interpolation. For each row, the left-most column and right-most column are respectively two images synthesized with two different latent codes. The remaining columns refer to images synthesized with interpolated latent codes.

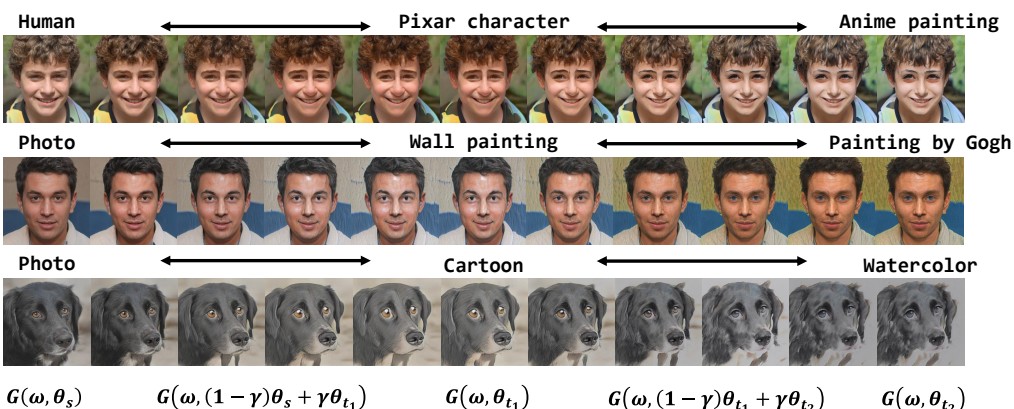

$$G(\omega, \theta_s) \quad G(\omega, (1-\gamma)\theta_s + \gamma\theta_{t_1}) \quad G(\omega, \theta_{t_1}) \quad G(\omega, (1-\gamma)\theta_{t_1} + \gamma\theta_{t_2}) \quad G(\omega, \theta_{t_2})$$

Figure 21: Cross-model interpolation. In each row, the left-most image is generated by the source generator. The middle and the right-most images are synthesized by two different target domain generators. The other images represent cross-model interpolations between two different domains.

### K.2 DIFFUSION-BASED IMAGE MANIPULATION

We implement based on Diffusion-CLIP (Kim et al., 2022) which seamlessly integrates with the existing zero-shot adaptation methods.

As illustrated in Fig 23, our method qualitatively exhibits a higher fidelity of target domain feature compared to previous methods. Quantitatively, our approach more closely aligns with the reference target images in CLIP space, indicating a greater semantic similarity.

Fig. 24 illustrates real-world image manipulation results for diffusion AIR.

## L USER STUDY

We conduct a user study to compare the quality and the diversity of the generated images with different schemes based on human feedback. The questionnaire is performed using the generated images by different schemes including NADA, IPL, SVL, and our proposed AIR. It includes 12 questions for quality evaluation and 4 questions for diversity assessment. We include examples for

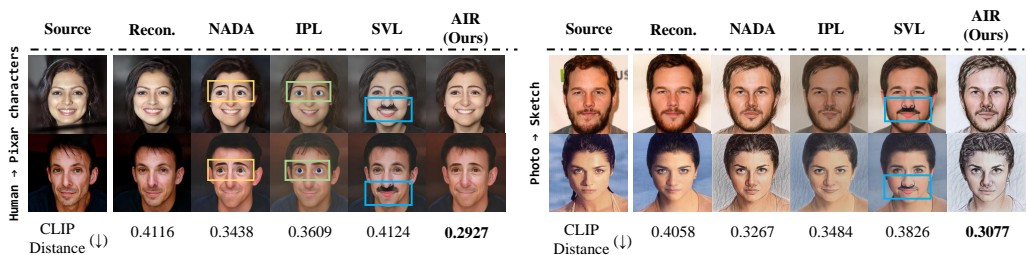

Figure 22: Image manipulation with GAN. The reference image are the same as in Fig. 1, 5.

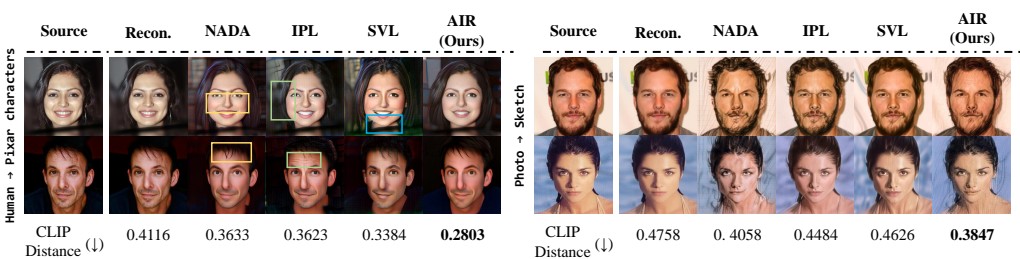

Figure 23: Diffusion model image manipulation. The reference images are the same as in Fig. 1, 6.

quality and diversity evaluation of our questionnaire in Fig. 25. Finally, we report the percentage of the user preference from 220 responses for each method and for both quality and diversity metrics in Tab. 3 in the main paper.

## M RELATED WORK

**Zero-shot Generative Model Adaptation** Zero-shot generative model adaptation is the task of adapting the source domain knowledge of a well-trained generator to the target domain without accessing any target samples. Unlike the zero-shot image editing methods (Patashnik et al., 2021; Shen & Zhou, 2021) where available modifications are constrained in the domain of the pre-trained generator, zero-shot generator adaptation can perform out-of-domain manipulation by directly optimizing the generator parameters. Previous works (Gal et al., 2022; Guo et al., 2023; Jeon et al., 2023) utilized the cross-modal representation in CLIP (Radford et al., 2021) to bypass the need for extensive data collection. Specifically, **NADA** (Gal et al., 2022) first proposes to use the embedding offset of textual description in the CLIP space to describe the difference between source and target domains. By assuming the text offset and image offset are well-aligned in CLIP space, it uses the text offset as adaptation direction and optimizes the trainable generator to align image offset with text offset. **IPL** (Guo et al., 2023) points out that adaptation directions in NADA for diverse image samples is computed from one pair of manually designed prompts, which will cause mode collapse, therefore they produce different adaptation directions for each sample. Similarly, **SVL** (Jeon et al., 2023) use embedding statistics (mean and variance) for producing adaptation direction instead of only mean of embeddings in NADA to prevent mode collapse.

However, the adaptation direction in previous work only focuses on the source and target domains and computes once before the generator adaptation. More importantly, all these methods assume the image and text offsets in the CLIP space are well aligned. In this paper, we draw inspiration from a similar problem called analogical reasoning in NLP, and empirically discover that the alignment of image and text offset in CLIP space is correlated to the concept proximity in CLIP space. Based on this finding, we proposed a method that iteratively updates the adaptation direction, which is more aligned with the image offset and more accurate for zero-shot adaptation with directional loss.

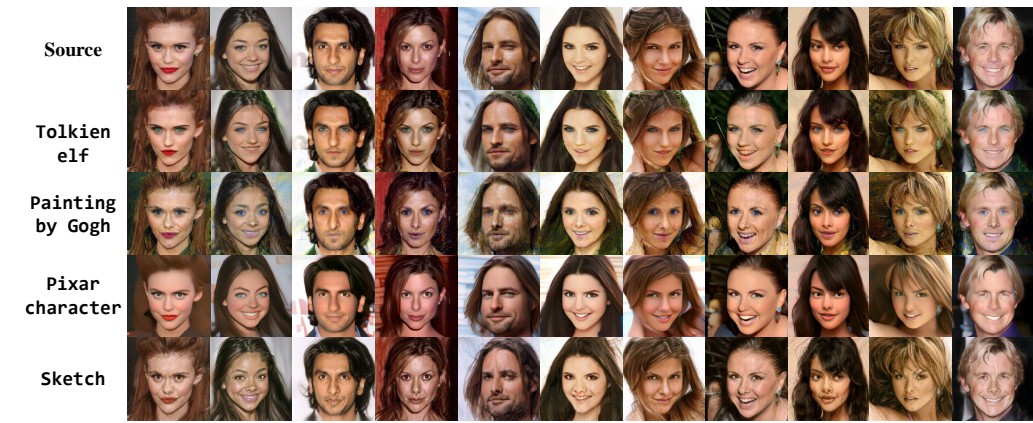

Figure 24: Additional results of image manipulation with diffusion model.

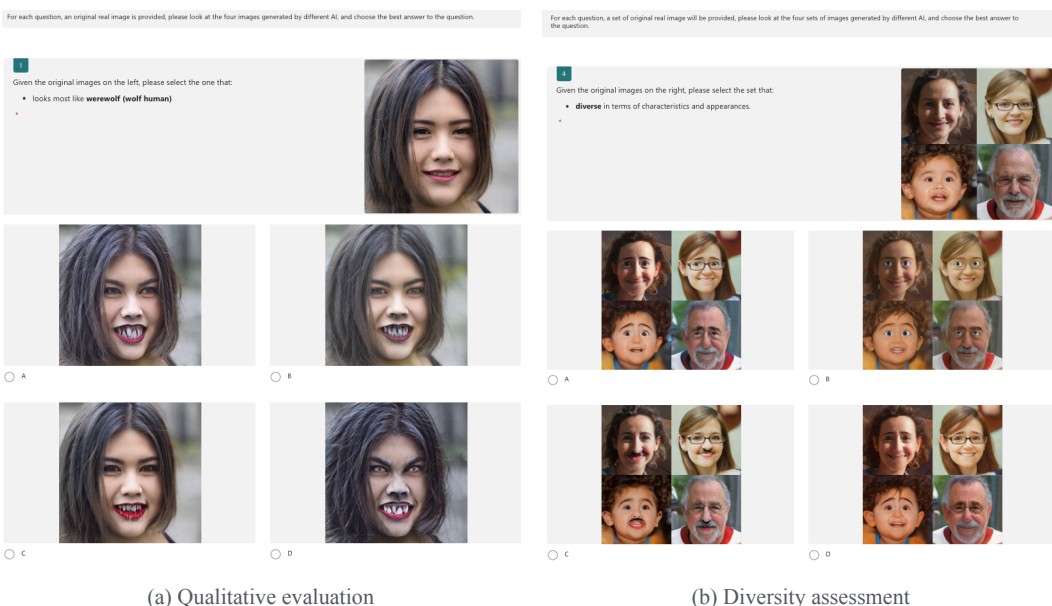

(a) Qualitative evaluation        (b) Diversity assessment

Figure 25: Examples of user study on (a) Quality assessment and (b) Diversity assessment.

**Analogical Reasoning** Research in NLP has shown that word representations of language models are surprisingly good at capturing semantic regularities in language (Collobert & Weston, 2008; Turian et al., 2010). Specifically, analogical reasoning (Mikolov et al., 2013c;a;b; Levy & Goldberg, 2014), utilizing the semantic regularities of word representations, aims to solve analogy tasks by using one pair of word vectors to identify the unknown member of a different pair of words, commonly via alignment of offsets, This is commonly modeled as using the vector offset between two words $a' - a$, and applying it to a new word $b$ to predict the missing word $b'$ that pair with $b$, as illustrated by the famous example of using $v(\text{"Man"}) - v(\text{"Woman"})$ and $v(\text{"King"})$ to identify $v(\text{"Queen"})$, where $v(\cdot)$ denotes word representation. This approach attracted a lot of attention for the vital role that analogical reasoning plays in human cognition for discovering new knowledge and understanding new concepts. It is already used in many downstream NLP tasks, such as splitting compounds (Daiber et al., 2015), semantic search (Cohen et al., 2015), cross-language relational search (Duc et al., 2015), etc.

Importantly, previous works (Levy et al., 2015; Köper et al., 2015; Vylomova et al., 2015) demonstrate that the effectiveness of analogical reasoning varies across different categories and semantic relations. More recent studies (Rogers et al., 2017; Fournier et al., 2020), present a series of experiments performed with BATS dataset (Gladkova et al., 2016) on various pre-trained vector space, e.g., GloVe

| Models | License |
|---|---|
| StyleGAN2 (Karras et al., 2020b) | Nvidia Source Code License |
| CLIP (Radford et al., 2021) | MIT License |
| StyleGAN2-pytorch (Karras et al., 2020b) | MIT License |
| e4e (Tov et al., 2021) | MIT License |
| StyleGAN-NADA (Gal et al., 2022) | MIT License |
| IPL (Guo et al., 2023) | MIT License |
| **Datasets** | **License** |
| FFHQ [5] | CC BY-NC-SA 4.0 |
| AFHQ [1] | CC BY NC 4.0 |

Table 12: Sources and licenses of the utilized models and datasets

(Pennington et al., 2014), Word2Vec (Mikolov et al., 2013b), and Skip-gram (Mikolov et al., 2013a), indicate that it is more effective to use $a' - a$ and $b$ to determine $b'$ when $b$ and $b'$ are close in vector space; and less so when $b$ and $b'$ are more apart.

Inspired by these studies, in this work, we perform an empirical study of offset misalignment in CLIP space and observe that for distant concepts in CLIP, image and text offset suffer from more misalignment, while closely related concepts suffer less. Based on our analysis, we proposed a method that iteratively refined the text offset for adaptation, which results in less offset misalignment and leads to a better generative model adaptation with directional loss.

# N LIMITATION

Our proposed iterative refinement method seeks to improve the quality of zero-shot adaptation. As noted by Guo et al. (2023), achieving adaptation across large domain gaps, such as Human to Cat, is particularly challenging. Similar to previous work, our approach necessitates that the trained generator somewhat closely approximates the target domain before initiating iterative refinement. Additionally, while our experiments on 32 different setups are comprehensive compared to previous work, more setups can be experimented to understand the limitations. We also note that our image-text offset alignment analysis focuses on CLIP-like multimodal representation spaces used in ZSGM work, and under ZSGM context.

# O SOCIAL IMPACT

The AIR methodology holds potential for enhancing artistic image synthesis in social media contexts and could serve as a beneficial data augmentation tool in other computer vision tasks such as recognition and detection. However, its capability to generate realistic images from real-world data raises ethical considerations. It is crucial to address these issues thoughtfully to prevent misuse and ensure responsible application of this technology.

# P USE OF LARGE LANGUAGE MODELS (LLMS)

LLMs were used solely as a writing aid to improve clarity, grammar, and style. They were not involved in generating research ideas, designing methodology, analyzing data, or drawing conclusions.

# Q LICENSES

In Table 12, we specify the source and licenses of the models and datasets used in our work. Note that the FFHQ dataset consists of facial images collected from Flickr, which are under permissive licenses for non-commercial purposes.

