# OpenReview forum: "AIR: Rethinking Image-Text Offset Alignment for Zero-Shot Generative Model Adaptation"
_ICLR.cc/2026/Conference — ICLR 2026 Conference Withdrawn Submission_

### Official Review · Reviewer_Ebxp · 2025-10-24

**Soundness:** 3
**Presentation:** 2
**Contribution:** 2
**Rating:** 4
**Confidence:** 3

**Summary:**

This paper investigates zero-shot generative model adaptation, which adapts pre-trained generative models using only text. It reveals that image-text offsets in CLIP space are misaligned and that this misalignment increases with concept distance. To address this, the authors propose Adaptation with Iterative Refinement, which iteratively refines text offsets and reduces misalignment through anchor sampling and prompt learning, aiming at improving sample quality.

**Strengths:**

- Zero-shot generative model adaptation is an interesting and valuable research direction.
- Investigating offset misalignment is practical and meaningful for improving ZSGM.
- Experimental results demonstrate performance gains in several scenarios.

**Weaknesses:**

- The stance of the paper appears somewhat confusing. The authors claim their motivation is to challenge the common assumption that text-image offsets are aligned, yet their proposed iterative method still relies on directional loss when selecting anchor samples, which implicitly assumes alignment. This reliance may reduce robustness — if the offset misalignment becomes severe and the image-text directions diverge significantly, the method’s effectiveness could be compromised.

- In several visual comparisons, the proposed method does not seem to outperform existing approaches and sometimes even appears slightly worse, such as in the bottom-right case of Fig. 1(c) and in Fig. 7 (Photo → Anime). These cases require clearer explanations or analysis.

- The paper does not clearly explain how the concept distance in Fig. 2 is computed.

- Fig. 4 is difficult to interpret due to the presence of many symbols and the lack of clear explanations in the caption describing what each symbol represents.

**Questions:**

Please refer to the weaknesses part above.

---

### Official Review · Reviewer_6tpz · 2025-10-27

**Soundness:** 3
**Presentation:** 3
**Contribution:** 3
**Rating:** 6
**Confidence:** 3

**Summary:**

The paper questions the key assumption behind all current zero-shot generative-model-adaptation (ZSGM) methods: that the geometric offset between two images in CLIP space is well aligned with the offset between their textual descriptions. Through extensive empirical analyses on six public datasets and four multimodal embedding spaces, the authors show that image–text offset alignment is often imperfect; the degree of mis-alignment grows as the semantic distance between the two concepts increases. Building on this observation, they introduce AIR (Adaptation with Iterative Refinement). After a few standard adaptation steps, the partially-adapted generator is treated as an “anchor”. AIR iteratively
• re-computes a new text offset towards the unknown target via a prompt-learning module, and
• continues adaptation with this refined offset.

Experiments on 32 settings (GANs and diffusion, rare concepts, artistic styles, etc.) show consistent quantitative, qualitative, and user-study gains over previous methods such as NADA, Mind-the-Gap, and Hap-DIT.

**Strengths:**

S1. Novel empirical finding
• First systematic study (to my knowledge) that measures image-text offset alignment inside CLIP for ZSGM.
• Demonstrates a clear and intuitively plausible correlation with concept distance.

S2. New algorithm motivated by the finding
• Iterative anchor sampling + prompt learning is simple yet effective.
• Focuses on quality improvement, a dimension often ignored in earlier work that prioritised word replacement or attribute transfer.

S3. Thorough evaluation
• 32 total setups, covering GANs & diffusion, multiple datasets, ablations, and a user study.
• Code promised in supplement; tables and figures appear reproducible.

S4. Potentially broad impact
• Suggests that many prior ZSGM papers rely on a shaky assumption; results may trigger re-examination of other “offset-based” paradigms.

**Weaknesses:**

W1. Limited theoretical explanation
The correlation is measured, but why distance causes mis-alignment is only speculated. A geometric or probabilistic model would strengthen the contribution.

W2. Anchor-selection hyper-parameters
AIR introduces several new knobs: number of iterations, anchor interval, prompt-learning LR, etc. Sensitivity analysis is not fully covered.

W3. Computational cost
Iterative adaptation means multiple forward/backward passes and several textual prompt-learning stages. Reported wall-clock time vs. previous methods is missing.

W4. CLIP-centric analysis
Although the paper tests four “contrastive multimodal spaces”, they are all CLIP derivatives (ViT-B/32, ViT-L/14, etc.). Would the same phenomenon hold for ALIGN, Florence or Kosmos? Could the method break if a future model exhibits smaller mis-alignment?

W5. User study details
The paper states “50 participants” but does not report demographic, interface, or statistical significance tests (e.g., p-value, CI).

**Questions:**

Q1. How sensitive is AIR to the prompt-length and vocabulary size used in the prompt-learning step?
Have you tried randomly initialising the prompt tokens versus initialising them with target-class names?

Q2. At each iteration you fine-tune both generator and prompts. Did you try freezing the generator and updating only prompts after the first anchor? How does quality change?

Q3. Runtime: on ImageNet-1K subset (“Rare”), what is total GPU-hour cost of AIR?

Q4. Could AIR be combined with “structure preservation” losses (e.g., DreamFusion’s CLIP-direction regulariser)? Do the improvements stack?

---

### Official Review · Reviewer_n8GK · 2025-10-28

**Soundness:** 3
**Presentation:** 3
**Contribution:** 2
**Rating:** 6
**Confidence:** 3

**Summary:**

The paper introduces Adaptation with Iterative Refinement (AIR), a novel framework for Zero-Shot Generative Model Adaptation (ZSGM). The primary contribution lies in the empirical discovery that image-text offset misalignment in CLIP space positively correlates with concept distance. Leveraging this, AIR proposes an iterative anchor sampling mechanism with a new prompt-learning strategy to dynamically infer text anchors, thereby refining the overall adaptation direction. Extensive experiments are conducted on both GANs and diffusion models to validate their adaptation quality.

**Strengths:**

1. The discovery of offset misalignment is sound and interesting.
2. The proposed method, AIR (Adaptation with Iterative Refinement), is clearly structured.
3. The authors present comprehensive experimental validation, spanning both GANs and diffusion models, including quantitative metrics, qualitative results, and user studies across many adaptation setups.

**Weaknesses:**

1. The author didn't justify the significance of ZSGM, as well as their proposed methods, clearly in the context of large-scale pre-trained VLM today. (See Q1 and Q2)
2. The paper’s foundational observation lacks necessary statistical validation, and the experimental justification for the core additive loss formulation and stability concerns is insufficient. (See Q3-Q6)

**Questions:**

1. Scope of $G_{\mathcal{S}}$: The experiments primarily utilize source generators pre-trained on narrow domains (FFHQ, AFHQ-Dog, LSUN-Church). The reported adaptation pairs (e.g., Human $\to$ Baby, Dog $\to$ Cat) do not strongly align with the claimed high-value applications of ZSGM in the introduction, “rare species, rare concepts, or artistic styles”.
2. $G_{\mathcal{S}}$ Choice. Please clarify the unique advantage of adapting these domain-specific generators via CLIP guidance in the experimental setting, rather than using generators pre-trained on large, diverse datasets, and conduct zero-shot adaptation on more challenging tasks.
3. Statistical validation is missing. The authors report a Spearman coefficient between concept distance and offset misalignment, but no *P-values* or significance test are provided.
4. Loss Design and Trade-off. Direction loss corresponds to global direction and adaptive loss to local precision, why not initialize $A_0 = S$ and rely solely on adaptive loss after a few iterations? In Sec 4.1, the authors state, “We empirically find that adding this adaptive loss to direction loss results in a more stable adaptation.” Should give more explanation or ablation studies.
5. Further, adaptive loss depends on dynamically learned text anchors $P_{\mathcal A_i}$ via prompt learning (Algorithm 2), which introduces approximation error. If it must needs direction loss as global guidance, given the trade-off, why is the loss set to ($\mathcal{L} = w_1 \mathcal{L}_{direction} + w_2 \mathcal{L}_{adaptive}$, $w_1 = w_2 = 1$) without exploring other weightings? Please report ablations for different weights or loss-only variants.
6. Instability from Early Activation. Table 9 shows degraded performance when $t_{\text{thresh}} $= 12.5% (or 25%) in the Human→Baby setting. The authors’ explanation (“neither excessively large nor small values …”) lacks depth. More detailed analysis of the instability and error accumulation introduced by early anchor use is needed.

---

### Official Review · Reviewer_L665 · 2025-11-01

**Soundness:** 2
**Presentation:** 1
**Contribution:** 3
**Rating:** 4
**Confidence:** 4

**Summary:**

This paper investigates a misalignment in CLIP-style contrastive learning models, identifying a consistent discrepancy between the text-image concept distance and the image-text score offset across various datasets. Based on this finding, the authors propose a Zero-Shot Generative Model Adaptation (ZSGM) framework called AIR. Instead of naively adapting based on CLIP score, AIR introduces an "anchor" — interpolated between source and target concepts — to facilitate smoother adaptation. The authors claim SoTA performance for AIR, supported by various quantitative, qualitative, and user study experiments.

**Strengths:**

- Valuable Insight (Misalignment): The paper's primary finding—that contrastive learning's objective does not explicitly enforce a metrically consistent relationship between semantic distance and score difference—is a valuable and non-trivial insight for the community, especially as these models are widely used as-is.

- Novel Adaptation Framework (AIR): Building on this insight, the proposal to use an "anchor" to guide adaptation from a source to a target concept is an intuitive and novel approach to addressing the identified misalignment.

- Comprehensive Validation: The authors have conducted a wide array of experiments, including quantitative metrics, qualitative examples, and a user study, to demonstrate the SoTA performance of their proposed AIR method.

**Weaknesses:**

While the core idea is promising, this paper is critically undermined by a disastrously poor presentation, which in turn makes its soundness difficult to verify. The paper is exceptionally difficult to read and verify.

1. Catastrophic Presentation and Readability:
  - Pervasive, Distracting Use of Boldface: The paper is littered with unprofessional and distracting bolding. The authors consistently emphasize useless meta-phrases (e.g., **"In this work," "Our primary findings"**) instead of the key technical terms or the findings themselves (e.g., **"misalignment," "score offset"**). This aggressive and poorly-judged emphasis makes the paper exceptionally difficult to read and follow.
  - Severe Formatting Violations: The submission appears to violate ICLR formatting guidelines. In-text citations and figure captions contain color (blue), which is reserved for the final, camera-ready version and is not permitted in the "under review" submission .
  - PDF Printing Disabled: The submitted PDF file appears to be locked against printing. This is a significant barrier to a thorough review, as many reviewers (myself included) rely on offline reading and annotation. This must be fixed.

2. Core Methodology is Incomprehensible (Definition Buried in Appendix): The central concept of the paper, the "anchor" is not defined in the 9-page main text. The critical definition—that the anchor is an "interpolation between tokenized source and target descriptions"—is improperly buried in Appendix D.2. According to ICLR guidelines, the main paper must be self-contained, and reviewers are not obligated to read the appendix.  Therefore, the paper as submitted is fundamentally incomplete. Its core methodology is unverifiable from the main text, making a thorough soundness evaluation impossible.


3. Insufficient User Study: The user study, which is presented as strong evidence of SoTA performance, is insufficiently documented. Appendix L contains no details on the number of participants, their expertise or demographics, or any mention of an IRB (Institutional Review Board) approval or ethical review process. As such, the results of this user study are unverifiable.

4. Minor Issues:
  - There appears to be a broken or incorrect reference in Figure 23.

**Questions:**

The paper is almost impossible to evaluate fairly due to the presentation issues. A successful rebuttal **must prioritize** clarity above all else.

1.  The "Anchor" Definition: Could the authors please provide a clear, concise, and easy-to-find definition of the "anchor"? How is it precisely formulated or set? This is the most critical question for understanding the paper.

2. User Study Details: Can the authors provide the missing details for the user study (participant count, demographics/expertise, and IRB approval)?

3. Formatting and Printing: Can the authors confirm that the formatting violations (color, PDF lock) will be fixed? What was the rationale for the pervasive misuse of boldface, which severely impacts readability?

### LLM Disclosure

I have used an LLM to assist with improving the grammar, clarity, and polishing of this review. The content, analysis, and final judgments are entirely my own.

---

### Note · Authors · 2025-11-14

I have read and agree with the venue's withdrawal policy on behalf of myself and my co-authors.